

# 1 Long-term surface energy balance of the western Greenland ice sheet

# 2 and the role of large-scale circulation variability

**Baojuan Huai[1], Michiel R. van den Broeke[2], Carleen H. Reijmer[2]**
1. College of Geography and Environment, Shandong Normal University, Jinan, China
2. Institute for Marine and Atmospheric Research, Utrecht University, Utrecht, The Netherlands
**Abstract.** We present the surface energy balance (SEB) of the west Greenland ice
sheet (GrIS), using an energy balance model forced with hourly observations from
nine automatic weather stations (AWS) along two transects: the K-transect with seven
AWS in the southwest and the T-transect with two AWS in the northwest. Modeled
and observed surface temperatures for non-melting conditions agree well, with
RMSEs of 1.1-1.6 K, while reasonable agreement is found between modeled and
observed 10-day cumulative ice melt. Absorbed shortwave radiation ($S_{net}$) is the main
energy source for melting (M), followed by the sensible heat flux ($Q_h$). The multi-year
average seasonal cycle of SEB components show that $S_{net}$ and M peak in July at all
AWS. The turbulent fluxes of sensible ($Q_h$) and latent heat ($Q_l$) decrease significantly
with elevation, and the latter becomes negative at higher elevations, partly offsetting
$Q_h$. Average June, July, August (JJA) albedo values are < 0.6 for stations below 1,000
m asl and > 0.7 for the higher stations. The near-surface climate variables and surface
energy fluxes from reanalysis products ERA-interim, ERA5 and the regional climate
model RACMO2.3 were compared to the AWS values. The newer ERA5 product
only significantly improves on ERA-interim for albedo. The regional model
RACMO2.3, which has higher resolution (5.5 km) and a dedicated snow/ice module,
unsurprisingly outperforms the re-analyses for (near-) surface climate variables, but
the reanalyses are indispensable to detect dependencies of west Greenland climate and
melt on large-scale circulation variability. We correlate ERA5 with the AWS data to
show a significant positive correlation of western GrIS summer surface temperature
and melt with the Greenland Blocking Index (GBI), and weaker and opposite
correlations with the North Atlantic Oscillation (NAO). This analysis may further help
to explain melting patterns in the western GrIS from the perspective of circulation
anomalies.

## 31 1 Introduction

In recent decades, the Greenland ice sheet (GrIS) has been a major contributor to
global sea-level rise, and is expected to remain so in the future (*Shepherd et al., 2019*),
raising worldwide concerns for coastal flooding and negative impacts on ecosystems
(*IPCC, 2019*). In-situ measurements provide crucial insights into the processes
causing temporal and spatial GrIS melt variability, notably how the various



components of the surface energy balance (SEB) contribute to snow and ice ablation. Automatic Weather Stations (AWS) monitor the near-surface atmospheric conditions on the ice sheet and -when equipped with radiation sensors- have proven to be excellent tools to determine the SEB and therewith quantify melt energy. At present there are >30 semi-permanent AWS installed on the GrIS. The largest GrIS AWS network currently operational is the Programme for Monitoring of the Greenland Ice Sheet (PROMICE; *Ahlstrøm et al., 2008; Van As et al., 2011*). PROMICE AWS are mainly situated in the narrow and low-lying ablation zone, and are operated by the Geological Survey of Denmark and Greenland (GEUS) in collaboration with the National Space Institute at the Technical University of Denmark (Greenland Survey). Other AWS networks are GC-Net, operated by the Cooperative Institute for Research in Environmental Sciences (CIRES; *Steffen and Box., 1996, 2001*), and situated mainly in the accumulation zone, and the K-Transect, a combined AWS-mass balance-ice velocity stake network operated since 1990 by the Institute for Marine and Atmospheric Research, Utrecht University (IMAU) (*Smeets et al., 2018*).

In recent decades, multiple observational studies described the local SEB on the GrIS. *Hoch et al. (2007)* made year-round radiative flux observations at Summit, the highest point on the GrIS. *Van den Broeke et al. (2008a, b)* and *Kuipers Munneke et al. (2018)* used measurements from four AWSs to describe the SEB along the K-transect in the southwestern GrIS. *Fausto et al (2016)* investigates two high melt episodes in the southern GrIS in the summer of 2012 and quantified and ranked melt energy sources through the melt season.

Until now, few studies addressed AWS- derived SEB and melt on the GrIS in terms of regional circulation variability. Statistical analysis suggests that southern GrIS climate responds strongly to atmospheric warming (*Hanna and Cappelen 2003*), and that Greenland overall has been one of the fastest warming regions of the Northern Hemisphere in the last 10~25 years (*Hanna et al., 2014*). These changes in GrIS summer near surface air temperature are caused both by changes in the local atmospheric heat balance and by changes in the large-scale atmospheric circulation (*Van den Broeke et al., 2017; Noël and others, 2019*). *Rajewicz and Marshall (2014)* state that "…circulation anomalies explain 38-49% of the summer air temperature and melt extent variability in GrIS over the period 1948-2013." Greenland high pressure blocking is a key feature of circulation variability in the western North Atlantic (*Ballinger et al., 2018*). Strong Greenland blocking episodes have been linked to exceptional surface melting of the western GrIS (*Hanna et al., 2014, Hanna et al. 2016*), and recently a Greenland Blocking Index (GBI) has been defined by *Fang (2004)* and *Hanna et al. (2013, 2014, 2015)*. Another important regional mode of large-scale atmospheric circulation variability is the North Atlantic Oscillation (NAO) (*Hurrell et al., 2003; Van den Broeke et al., 2017*).

To study the dependency of regional west Greenland SEB and melt on large-scale circulation variability we use data from two GrIS AWS transects, i.e. the southwestern Kangerlussuaq (K-) transect and the northwestern Thule (T-) transect. In addition, we use reanalysis (ERA5, ERA-Interim) data and output of a regional





atmospheric climate model (RACMO2.3) to obtain spatially continuous results. ERA5
is the latest reanalysis product from the European Centre for Medium-Range Weather
Forecasts (*ECMWF; Dee et al., 2011; Hersbach and Dee, 2016*), and replaces
ERA-Interim, considered to be the leading product over GrIS until now (*Albergel et
al., 2018; Bromwich et al., 2016*). However, so far little is known about the
performance of ERA5 over the GrIS. Because both the PROMICE and IMAU AWS
are not  assimilated in ERA5, these data can be used to assess its quality and that of
regional climate models. Thus, we also include an evaluation of ERA5/RACMO2.3
SEB components over the western GrIS.
This paper is organized as follows. The AWS sites and data used to force the
SEB model are described in Section 2, followed by the SEB model description in
Section 3. The results Section 4 is split into three parts: we present the SEB results
along the two GrIS transects and we evaluate the near-surface climate and SEB in
ERA5 and RACMO2.3, after which we discuss their dependency on the large-scale
circulation indices GBI and NAO.

**2   Study sites, observational and model data**
**2.1 AWS transects**
To calculate the SEB and melt rate, we use data of all seven AWS along the
K-transect in the southwestern GrIS, i.e. four IMAU AWS (S5, S6, S9 and S10) and
three PROMICE AWS (KAN_L, KAN_M and KAN_U, Fig. 1c). We also use data of
the two PROMICE AWS located near Thule, dubbed the T-transect, in the
northwestern GrIS (THU_L and THU_U, Fig. 1b). The K-transect was initiated in the
summer of 1990 as part of the Greenland Ice Margin EXperiment (*GIMEX;
Oerlemans and Vugts 1993; Kuipers Munneke et al., 2018*) and originally represented
an array of three AWS (S10 was added later) and eight surface mass balance/ice
velocity sites. In 2008 and 2009, three more sites were added to the K-transect as part
of the PROMICE AWS network (*Van As et al., 2011; Fausto et al., 2012a*). The
topographic details as well as the observational period, climate characteristics and
AWS sensor specifications are listed in Tables 1 and 2.





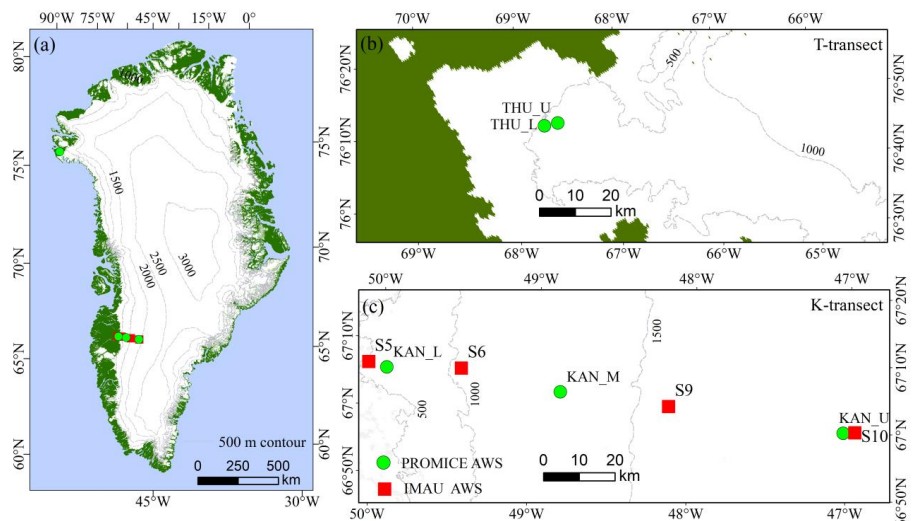


**Fig 1.** The two GrIS AWS transects used in this study (a): blue represents ocean, green ice-free
tundra and white glaciated areas and location of AWS sites. The transects are magnified in b) and
c). Red squares are IMAU AWS and green circles PROMICE AWS. Grey dashed lines are 500 m
elevation contours.

**Table 1** AWS location, elevation and start of observations

| Station | Latitude(N) | Longitude(W) | ELA(m a.s.l) | Start Date |
|---------|-------------|--------------|--------------|------------|
| S5 | 67.08 | 50.10 | 490 | 27/08/2003 |
| S6 | 67.07 | 49.38 | 1020 | 01/01/2003 |
| S9 | 67.05 | 48.22 | 1520 | 26/08/2003 |
| S10 | 67.00 | 47.02 | 1850 | 17/08/2010 |
| KAN_L | 67.10 | 49.95 | 670 | 01/09/2008 |
| KAN_M | 67.07 | 48.84 | 1270 | 02/09/2008 |
| KAN_U | 67.00 | 47.03 | 1840 | 04/04/2009 |
| THU_L | 76.40 | 68.27 | 570 | 09/08/2010 |
| THU_U | 76.42 | 68.15 | 760 | 09/08/2010 |

**Table 2** AWS sensor specifications

| Sensors | PROMICE Type | IMAU Type | PROMICE Accuracy | IMAU Accuracy |
|---------|--------------|-----------|------------------|---------------|
| Temperature | MP100H-4-1-03-00-10DIN | Vaisala HMP45C | < 0.1 K | 0.4°C at −20°C |
| Air pressure | CS100-Setra model 278 | Vaisala PTB101B | 1.5 hPa | 4 hPa |
| Wind speed | 05103-5. R.M. Young | 05103-L.R.M.Young | $0.3 \text{ m s}^{-1}$ | $0.3 \text{ m s}^{-1}$ |
| Wind direction | 05103-5.R.M. Young | 05103-L.R.M.Young | 3 ° | 3 ° |





| | | | | |
|---|---|---|---|---|
| Humidity | HygroClip S3 | Vaisala HMP45C | 1.5 % RH/0.3 ℃ | 2% for RH <90% |
| Radiation | Kipp Zonen CNR1 or CNR4 | Kipp Zonen CNR1 | 10% of daily totals | 10% of daily totals |
| Surface height | SR50A sonic ranger | SR50 sonic ranger | 1 cm or ±0.4%[*] | 0.01 m |
| | Ørum & Jensen NT1400 pressure transducer | | 2.5 cm[*] | |

*PROMICE AWS pressure transducer sensor accuracy from Fausto et al. (2012)
**2.2 Data**
2.2.1 AWS data and processing
Hourly average wind speed, incoming and reflected shortwave radiation,
incoming and emitted longwave radiation, air temperature, relative humidity and air
pressure are used to drive the SEB model. To illustrate the data time series at the nine
AWS, Figure 2 shows the full temperature records, where temperature is recalculated
to the reference height of 2 m using the SEB model. Note that S6 data gaps include
large parts of 2008, 2010, 2012 and 2015, while the other AWS have generally more
complete coverage.
Snow and ice height records cannot always be used directly to assess sensor
height changes because of AWS design changes and/or settling of the structure. For
PROMICE AWS, we use the results from a physically based method to remove
air-pressure variability from the signal of the pressure transducer records (*Fausto et
al., 2012b; Van As et al., 2011*). For details of S5, S6, S9 and S10 data biases,
corrections, and data gap filling in case of sensor failure, we refer to *Smeets et al.*
*(2018)*.
In the model evaluation using surface temperature, we use all-period data, while
in the evaluation of melt using measured height changes, we use the period starting in
2008 to maximize data overlap. For annual or multi-year averages of SEB
components, we use complete years only (Table 3 and Table S1 and Figures 7, 8, 9
and 10). Data points used in Figure 4 coincide with time series in Figure 2, while
Figure 6 starts in 2008.


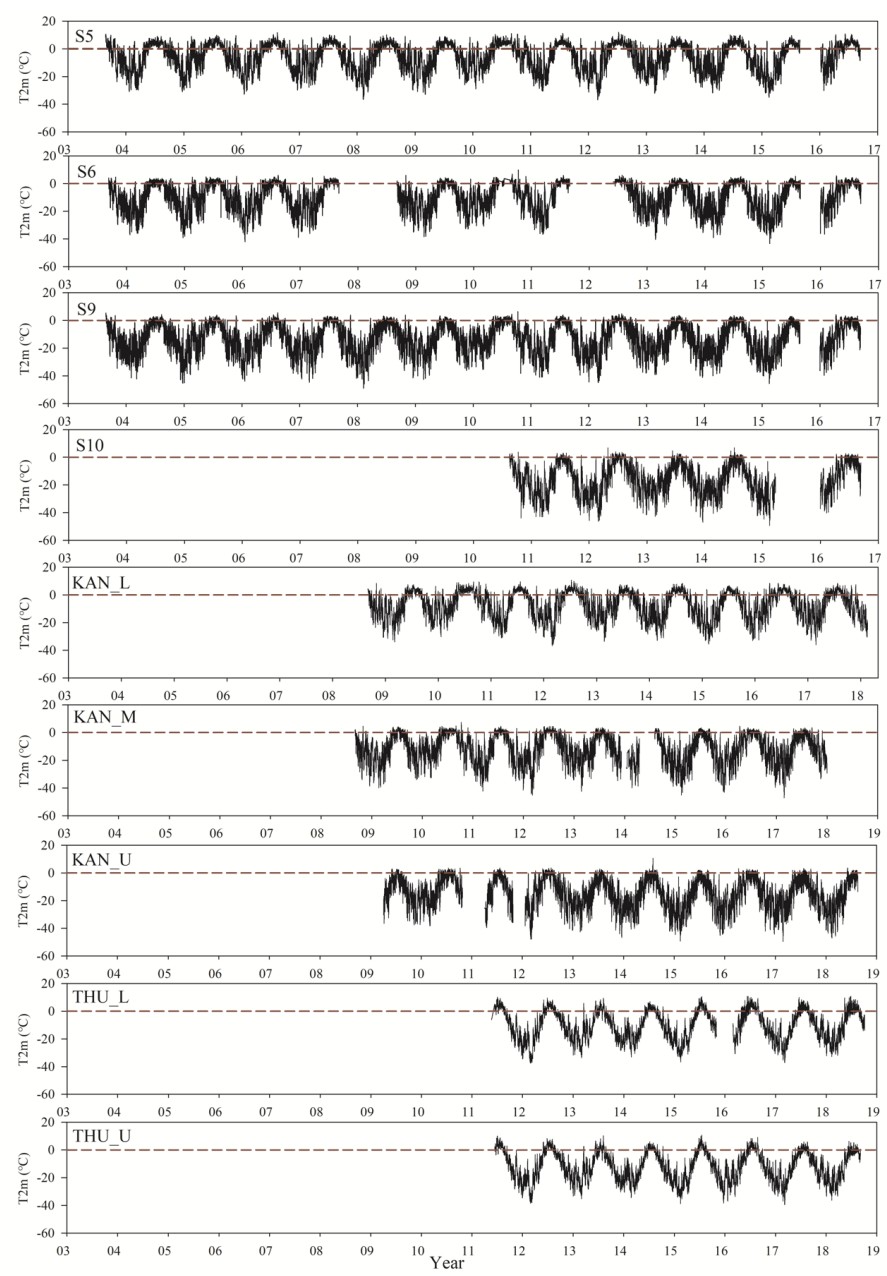

**Fig 2.** Time series of 2 m temperature (T2m) at the nine AWS sites used in this study

2.2.2 ERA5

The fourth-generation European Centre for Medium Range Weather Forecasts (ECMWF) Interim Reanalysis (*ERA-Interim, Dee et al., 2011*), available at a spatial





resolution of 0.75 °and a 6-hourly time resolution, has been widely used over the GrIS
(*Bromwich et al., 2016, Albergel et al., 2018*). ERA-Interim is not continued beyond
August 2019, and is replaced by the follow-on product ERA5. The latter has a higher
spatial (31 km) and temporal (hourly) resolution (*ECMWF, 2018; Delhasse et al,*
*2019*). Beside the higher time and horizontal resolution and updated physics package,
the main improvements for ERA5 compared to ERA-Interim are a higher number of
vertical levels, an improved 4D-VAR assimilation system and more data assimilated
(*ECMWF, 2018*). In addition to using ERA5 near-surface climate variables and SEB
components for evaluation, we also use ERA5 500 hPa geopotential height for the
GBI and NAO regression analysis.
2.2.3 RACMO2.3

The Regional Atmospheric Climate Model (RACMO2) is developed and
maintained at the Royal Netherlands Meteorological Institute (KNMI) (*Van Meijgaard*
*et al., 2008*). The polar version of RACMO2 was developed at IMAU, to specifically
represent the SMB of polar ice sheets such as the GrIS (*Ettema et al., 2010*).
RACMO2.3 incorporates the dynamical core of the High-Resolution Limited Area
Model and the physics from the ECMWF Integrated Forecast System (*ECMWF-IFS.,*
*2008; Noël et al., 2018*). We use output at 5.5 km horizontal spatial resolution of the
polar version of RACMO2.3 for the period 2003-2018 with a daily time resolution
*(Noël et al., 2018)* for evaluation and monthly 2 m temperature and melt flux data for
GBI and NAO correlation analysis presented in Section 2.2.4.
2.2.4 Monthly GBI and NAO index

The Greenland Blocking Index (GBI) represents the mean 500 hPa geopotential
height for the 60-80 °N, 20-80 °W region (*Hanna et al., 2014, 2015*), while the North
Atlantic Oscillation (NAO) index represents the normalized sea level pressure
difference between Iceland and the Azores (*Hurrell et al., 1995; Jones et al., 2003;*
*Hurrell et al., 2012*). The GBI and NAO-index time series are made available by the
US National Oceanographic and Atmospheric Administration (NOAA)'s Earth
System Research Laboratory Physical Sciences Division at: http://www.esrl.noaa.gov/
psd/data and are plotted in Figure 3, in which the blue and red dots represent June,
July and August (JJA) values. The two indices are not independent, with a correlation
coefficient between JJA NAO and GBI values for this period of -0.65, i.e. Greenland
blocking is associated with less zonally oriented large-scale flow over the North
Atlantic, as expected.



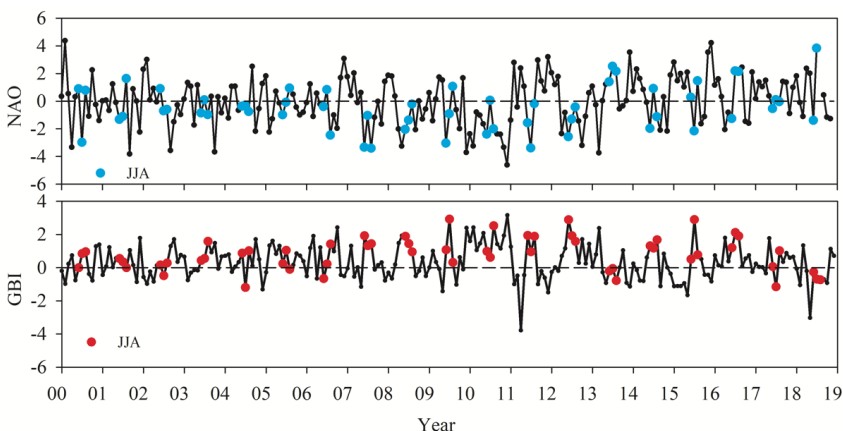


**Fig 3**. Time series of monthly average NAO and GBI indices where the blue and red dots are values for June, July, August (JJA)


## 3 Surface energy balance model

### 3.1 Model description

The Surface Energy Balance (SEB) model uses AWS data as input and solves for the surface temperature $T_s$ at which the SEB is closed, i.e.:

$$M=S_{in}+S_{out}+L_{in}+L_{out}+Q_h+Q_l+G+Q_p \qquad (1)$$

in which $M$ is the energy used for melt ($M = 0$ when $T_s < 273.15$K), $S_{in}$ and $S_{out}$ are the observed incoming and reflected shortwave radiation fluxes, $L_{in}$ and $L_{out}$ are the observed incoming and calculated outgoing longwave radiation fluxes (assuming unit emissivity), $Q_h$ and $Q_l$ are the calculated sensible and latent turbulent heat fluxes, $G$ is the surface value of the calculated sub-surface heat flux and $Q_p$ is the heat flux supplied by rain. All fluxes are evaluated at the surface and fluxes towards the surface are defined positive. In this study, $Q_p$ is neglected because no information on rainfall timing and rate is available. A previous study used precipitation data from the HIRHAM5 regional climate model bi-linearly interpolated to AWS locations, and reported that the rain heat flux on average contributed ~1% to the melt flux in summer at the southern GrIS sites QAS_L (*Fausto et al., 2016*).

$Q_h$ and $Q_l$ are estimated using the bulk aerodynamic approach with stability corrections based on Monin-Obukhov similarity theory (*Van den Broeke et al., 2005; Smeets and Van den Broeke., 2008*), using the stability functions of *Holtslag and de Bruin., 1988*. The expressions used to calculate $Q_h$ and $Q_l$ are as follows:

$$Q_h =\rho_\alpha c_p\, u_*\, \theta_* =\rho_\alpha c_p C_H u(\theta-\theta_s) \quad (2)$$

$$Q_l =\rho_\alpha L_v\, u_*\, q_* =\rho_\alpha L_v C_E u(q-q_s) \quad (3)$$





Where $u_*$, $\theta_*$ and $q_*$ are the turbulent scales for momentum, heat and moisture, $c_p$
is the specific heat capacity of air at constant pressure, $\rho_a$ is air density, $L_v$ is the latent
heat of sublimation and $C_H$ and $C_E$ are bulk exchange coefficients for heat and
moisture, respectively. The SEB model uses the measured atmospheric temperature,
wind speed and humidity at the AWS sensor level together with the (iteratively
estimated) surface temperature, assuming zero wind speed and saturated humidity
values at the surface. The surface roughness length for momentum ($z_0$) varies strongly
in time and space in the ablation zone of GrIS, and is often set to different constant
values for snow and ice surfaces *(Smeets and van den Broeke., 2008; Brock et al.,*
*2006)*, while the values for heat ($z_h$) and moisture ($z_q$) are estimated following the
expressions due to *Andreas et al. (1987)*. Based on previous work with observations
from both the lower and upper measurement levels to compute a temporally evolving
$z_0$ value at sites S5 and S6 *(Smeets and van den Broeke., 2008)*, a $z_0$ value of $1.3 *10^{-3}$
m is chosen for S5, S6, and KAN_L when ice is at the surface, and $1.3* 10^{-4}$ m when
snow covers the surface at these AWS sites. At S9, S10, KAN_M and KAN_U, we
use a constant $z_0$ value of $1 *10^{-3}$ m for ice as the annual cycle is much smaller at
these stations *(Van den Broeke et al., 2005)*, while $1 *10^{-4}$ m is used for snow. At
THU_L and THU_U, we use ice values of $1.2 *10^{-3}$ m and $1 *10^{-3}$ m and snow values
of $1.3 *10^{-4}$ m and $1 *10^{-4}$ m for THU_U, respectively. The $z_0$ values of all the stations
are listed in Tables 3.
**Table 3**   The surface roughness length for momentum ($z_0$) at the nine AWS sites

| Station | Ice $z_0$ | Snow $z_0$ |
|---------|-----------|------------|
| S5 | $1.3 *10^{-3}$ | $1.3 *10^{-4}$ |
| S6 | $1.3 *10^{-3}$ | $1.3 *10^{-4}$ |
| S9 | $1.0 *10^{-3}$ | $1.0 *10^{-4}$ |
| S10 | $1.0 *10^{-3}$ | $1.0 *10^{-4}$ |
| KAN_L | $1.3 *10^{-3}$ | $1.3 *10^{-4}$ |
| KAN_M | $1.0 *10^{-3}$ | $1.0 *10^{-4}$ |
| KAN_U | $1.0 *10^{-3}$ | $1.0 *10^{-4}$ |
| THU_L | $1.2 *10^{-3}$ | $1.3 *10^{-4}$ |
| THU_U | $1.0 *10^{-3}$ | $1.0 *10^{-4}$ |

The *G* calculation uses the vertical temperature distribution in the near surface
snow layers, as calculated in the sub-surface part of the SEB model, based on the
SOMARS model (Simulation Of glacier surface Mass balance And Related
Sub-surface processes, *Greuell and Konzelman, 1994*) with skin layer formulation
*(Van den Broeke et al., 2011)* in which penetration of shortwave radiation is neglected
*(Van den Broeke et al., 2011)*. For a more detailed description of the model and recent
applications, we refer to *Reijmer (2002, 2008)*, *Van den Broeke (2004, 2008a,b, 2011)*,
*Kuipers Munneke (2009, 2012, 2018)*.





## 3.2 SEB model evaluation

The calculation proceeds as follows. The SEB components $L_{out}$, $Q_h$, $Q_l$ and $Q_g$ are expressed in terms of surface temperature, and the SEB model then iteratively searches for the value of $T_s$ at which the SEB is closed. When $T_s$ exceeds the melting point, it is set to 273.15 K and the remaining energy is used for melting. The root-mean-square-error (RMSE) between hourly modelled and observed $T_s$, the latter derived from $L_{out}$ assuming unit emissivity, is used to evaluate model performance at the nine AWS locations in Figure 4. The RMSE varies from 1.1 K at KAN_U to 1.6 K at S10. The results show that at KAN_M (RMSE=1.1), KAN_U (RMSE=1.1), THU_L (RMSE=1.2) and THU_U (RMSE = 1.1) the model performs better than at S5 (RMSE=1.6) and S10 (RMSE =1.6). Overall, at the 9 AWS, observed and modeled surface temperatures agree largely to within the observational uncertainty.

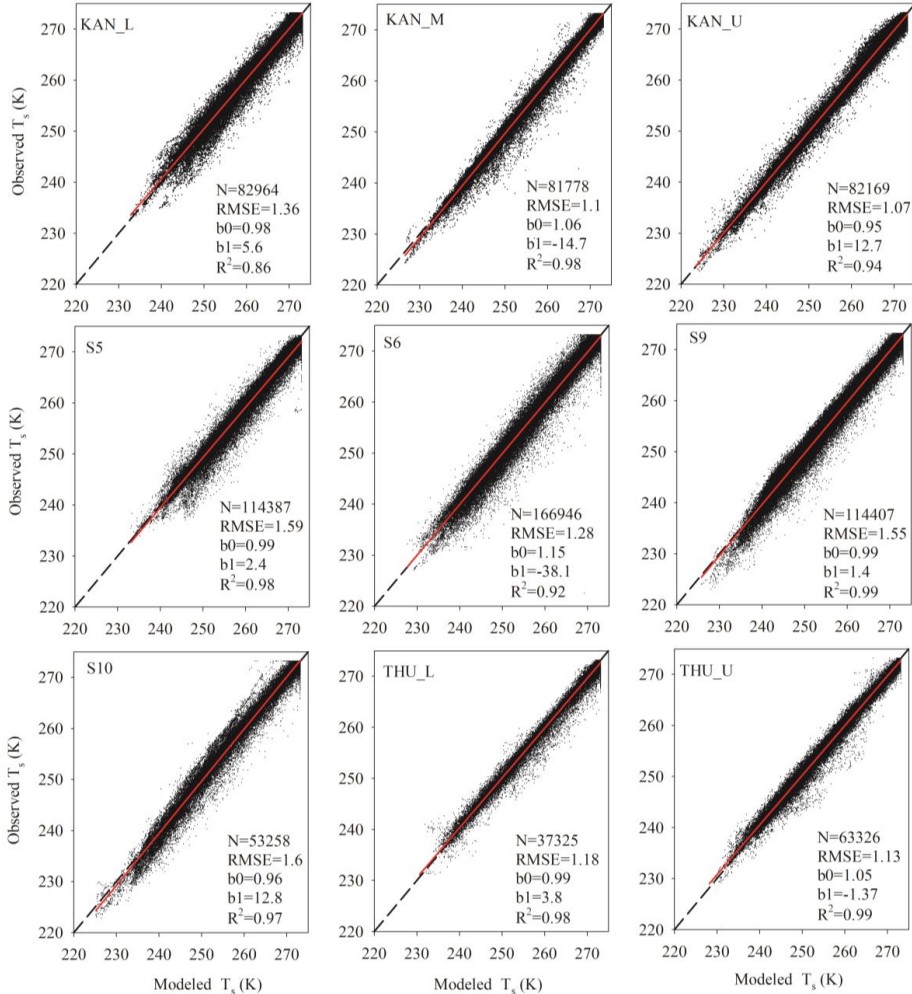

**Fig 4**. Modeled and observed hourly surface temperature $T_s$ for the nine AWS. The dashed black line represents the 1:1 line and the red solid line the linear regression. Statistics show the number of data points (N), root-mean-squared-error (RMSE), regression slope ($b_0$) and intercept ($b_1$), and coefficient of determination ($R^2$).

During melt, when the surface temperature is fixed at 273.15 K, $T_s$ can no longer be used for evaluation, and we assess model performance by comparing observed and modeled ice melt, assuming the density of ice to be known. This does not work for S9, S10 and THU_U which are situated above the equilibrium line, and hence on firn with unknown density. A 10-day period is chosen, to reduce the measurement noise so that a meaningful comparison is possible (*Van den Broeke et al., 2008b*). The corrected pressure transducer melt data collected by PROMICE AWS and SR50A sonic ranger collected by IMAU AWS are converted to mass changes (mm w.e.) by assuming an ice density of 910 kg/m³. Figure 5 shows reasonable agreement between modeled and observed 10-day ice melt for KAN_L, KAN_M, S5, S6 and THU_L.

At S5 and S6, *Van den Broeke et al.* (2008b) and *Kuipers Munneke et al.* (2018) compared annual ice ablation versus stake observations. They found that although results agreed within the model and measurement uncertainty, the relative differences for individual years could be substantial, up to 20%. Here, differences for individual 10-day periods of up to 46% are found, but the average difference is small, 6%.

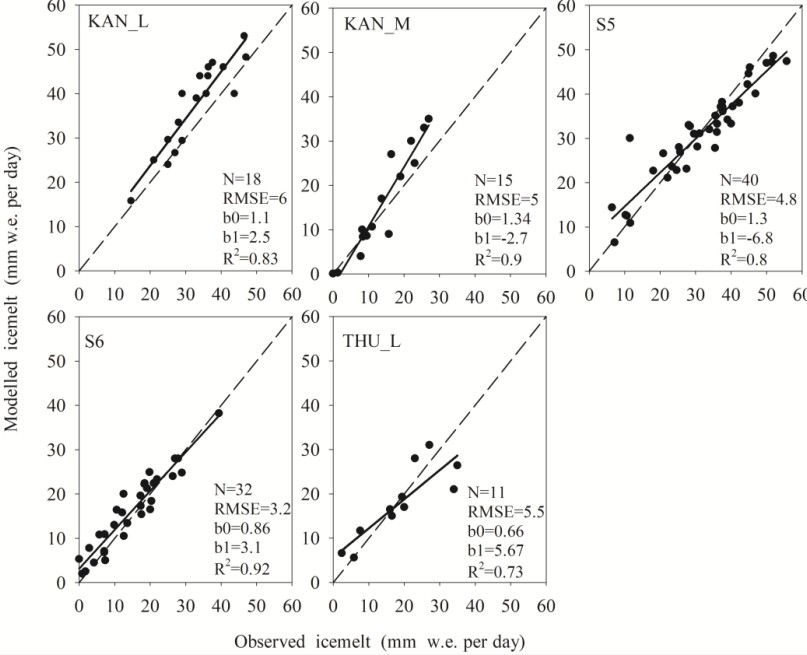

**Fig 5**. Average 10-day modeled and observed ice melt (expressed in mm w.e. per day) for the five AWS situated in the ablation zone, assuming an ice density of 910 kg/m³. The dashed line is the 1:1 line and the solid line the linear regression line. Statistics show the number of data points



(N), root-mean-squared-error (RMSE), regression slope ($b_0$) and intercept ($b_1$), and coefficient of
determination ($R^2$).
Apart from model uncertainties, there are various possible explanations for the
differences. *Fausto et al (2016)* show that in the lower ablation area in the southern
GrIS (QAS_L), the average rain energy flux in JJA averaged 1% of the total melt
energy flux but can reach 5 - 9 % during high melt episodes. *Van den Broeke et al*
*(2008b)* calculated the impact of radiation penetration on melt at S5, and showed that
melt energy was only slightly smaller than in the case without radiation penetration.
Based on this we expect that neglecting subsurface radiation penetration has little
effect on the total cumulative melt flux.
**4 Results and Discussion**
**4.1 SEB and comparison of the two transects**
4.1.1 Surface height change
The measured surface height change and modelled cumulative ice melt for the
seven K-transect stations (S5, S6, S9, S10 and KAN_L, KAN_M, KAN_U) are
shown in Figure 6. From 2008 to 2017, the ablation at S5 and KAN_L reached nearly
30 m of ice while for the stations above the equilibrium line (~1500 m a.s.l.) the total
accumulation was about 4 m of firn. At site S5 (490 m a.s.l.) the modeled ice melt and
measured surface height change agree well, even in winter, indicating that there is
little snow accumulation in winter at this site, as supported by visual observations. At
site KAN_L (670 m a.s.l.), there are obvious accumulation events in the winter in
2009 and 2011, and modeled ice melt is generally larger than observed. The strongest
melt occurred in summer 2012, contributing to the largest annual ice-sheet mass loss
on record (*Khan et al., 2015*), followed by a return to more average conditions in
2013 (*Nghiem et al., 2012; Kuipers Munneke et al., 2018*). Overall, modelled and
observed total height change agree typically within 10%.



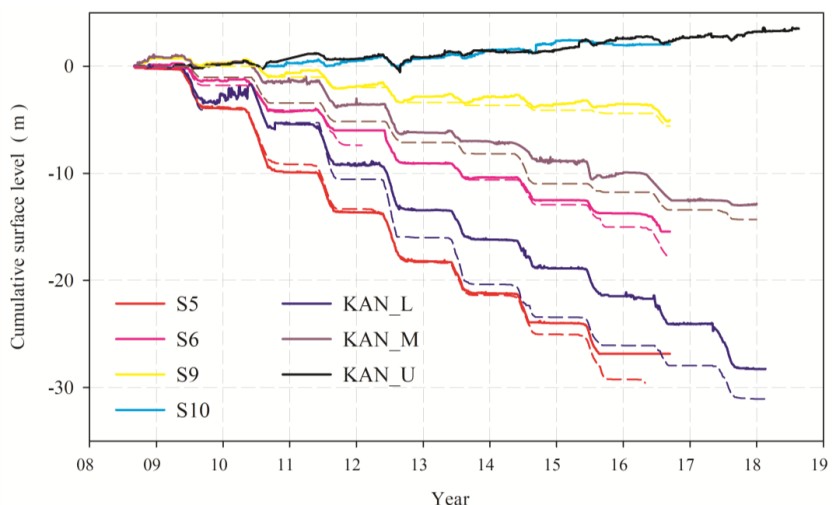


**Fig 6.** Measured height changes (solid lines) and modelled ice melt (dashed line) at the seven
K-transect AWS.

### 4.1.2 SEB components

Table 4 shows that average summer (June, July, August; JJA) net shortwave
radiation $S_{net}$ provides most (67% at S5 to 95% at S9) of the energy used for heating
or melting the surface along both transects (*Van As et al., 2012; Van den Broeke et al.,
2008b; 2009*). On average, $S_{net}$ is largest at KAN_L (125 W m$^{-2}$), and smallest at S10
(65 W m$^{-2}$). For the T-transect, average $S_{net}$ decreases from 84 W m$^{-2}$ at THU_L to 74
W m$^{-2}$ at THU_U. The generally lower values in the northwestern GrIS can be
explained by the difference in latitude but also by a smaller value of the shortwave
transmissivity (0.63 at KAN_L vs. 0.53 at THU_L in summer, using top-of
-atmosphere radiation data from ERA5), probably owing to more frequent and thicker
clouds along the T-transect (cloud cover 0.51 at KAN_L vs. 0.56 at THU_L in
summer, using cloud cover estimates from PROMICE AWS based on $L_{in}$ and air
temperature). Along the K-transect, JJA $L_{in}$ ranges between 250 and 285 W m$^{-2}$, while
$L_{out}$ varies between 298 and 314 W m$^{-2}$. Along the T-transect, $L_{in}$ is 273 to 279 W m$^{-2}$
and $L_{out}$ 309 to 312 W m$^{-2}$. The reduced longwave heat loss confirms higher
cloudiness in the northwest GrIS, in agreement with *Van As et al. (2012)*.

**Table 4** Energy fluxes (W m$^{-2}$) averaged over June, July, August (JJA) at the nine AWS
locations, SEB values of $L_{out}$, $Q_h$, $Q_l$, G and M are derived from the SEB model while $S_{in}$, $S_{out}$ and
$L_{in}$ are from observations.

| Flux | S5 | KAN_L | S6 | KAN_M | S9 | KAN_U | S10 | THU_L | THU_U |
|---|---|---|---|---|---|---|---|---|---|
| $S_{in}$ | 249 | 265 | 267 | 256 | 294 | 300 | 295 | 231 | 249 |
| $S_{out}$ | -136 | -140 | -154 | -158 | -211 | -234 | -230 | -147 | -176 |
| $S_{net}$ | 113 | 125 | 113 | 114 | 92 | 66 | 65 | 84 | 74 |





| | | | | | | | | | |
|---|---|---|---|---|---|---|---|---|---|
| $L_{in}$ | 285 | 283 | 265 | 263 | 255 | 250 | 253 | 279 | 273 |
| $L_{out}$ | -314 | -314 | -311 | -308 | -306 | -298 | -301 | -312 | -309 |
| $L_{net}$ | -29 | -29 | -45 | -44 | -51 | -48 | -48 | -33 | -36 |
| $R_{net}$ | 84 | 96 | 68 | 70 | 42 | 18 | 17 | 51 | 38 |
| $Q_h$ | 45 | 28 | 18 | 8 | 5 | 7 | 3 | 21 | 11 |
| $Q_l$ | 3 | -3 | -0.4 | -10 | -4 | -12 | -6 | -11 | -6 |
| G | -10 | 1 | -8 | -6 | -0.2 | 7 | 7 | 1 | 1 |
| M | -122 | -119 | -77 | -62 | -42 | -20 | -20 | -61 | -44 |

$S_{net}$ clearly is the main energy source for heating and melt at the ice sheet in
summer, followed by the sensible heat flux. $Q_h$ is larger than $Q_l$ for the low elevation
stations, with a JJA average of 45, 28 and 18 W m$^{-2}$ for S5, KAN_L and S6,
respectively, indicating significant contributions to the melt energy. At higher
elevations, $Q_h$ becomes small and $Q_l$ significantly negative (sublimation), with a JJA
average of -4, -12 and -6 W m$^{-2}$ for S9, KAN_U and S10, respectively. As a result,
above the equilibrium line, the two turbulent fluxes tend to (partly) cancel. However,
summertime $S_{net}$ and $L_{net}$ are also negatively correlated, indicating that net radiation
$R_{net}$ is always substantially smaller than $S_{net}$. This means that, when compared to $R_{net}$,
$Q_h$ does provide a significant contribution to summer melt and surface heating energy,
ranging from 12% at S9 to 37% at S5.
The important role of $Q_h$ in the GrIS SEB becomes even more evident if we look
at annual mean SEB components (Table S1 in the Supplementary Materials). In winter,
$Q_h$ becomes the main source of surface warming. In the absence of absorbed
shortwave radiation, wintertime $Q_h$ balances a large part of $L_{net}$ so that annual mean
$Q_h$ is relatively large and annual $R_{net}$ at S5, KAN_L, S6 and KAN_M becomes small
with values of 10, 14, 23 and 6 W m$^{-2}$, respectively, and even becomes negative for
the higher stations S9, KAN_U and S10. Sites with negative annual mean $R_{net}$ are very
rare at the Earth's surface, and require an efficient local atmospheric heat source,
which over the GrIS is provided by the mixing of relatively warm air aloft to the ice
sheet surface by katabatic winds, resulting in large $Q_h$ and large negative $L_{out}$. Annual
average values of $Q_h$ are as high as 32 W m$^{-2}$ for S5 decreasing to 6 W m$^{-2}$ at S10, 20
W m$^{-2}$ for THU_L and 16 W m$^{-2}$ for THU_U. The annual mean latent heat flux $Q_l$
varies between -1 W m$^{-2}$ and -6 W m$^{-2}$.
Figure 7 shows the interannual variability of the annual melt energy and the
corresponding melt water equivalent. The legend lists the percentage contribution
from JJA melt for each station. Significant inter-annual variability is present in the
annual melt energy; the standard deviation of the annual melt as a fraction of the
average value for stations with > 5 years of data ranges from 119 MJ m$^{-2}$ (61% of the
mean) at KAN_U to 209 MJ m$^{-2}$ (39%) at KAN_M. For most locations, 2010 and/or
2012 were the strongest melt years, with the highest ablation of 4.8 m w. e. per year
being reached at S5 in 2010. Only S5 (85%) and KAN_L (84%) experience
significant (>10%) non-summer melt, otherwise JJA melt energy contributes more





than 90% to the annual total melt energy. No significant trend is present in any of
these time series, because they are all relatively short and exhibit large year-to-year
variability.

Melt (M) at the K- transect AWS sites is significantly higher than at the T-
transect: average annual magnitude of M for THU_L is 512 MJ m$^{-2}$ compared to 1160
MJ m$^{-2}$ and 1133 MJ m$^{-2}$ for S5 and KAN_L, respectively. Obviously, this can be
partly explained by differences in absorbed short-wave radiation caused by the
different latitudes of the two transects and the lower temperatures further north,
resulting in a shorter ablation season. In the discussion section, we address the
potential role of atmospheric circulation.

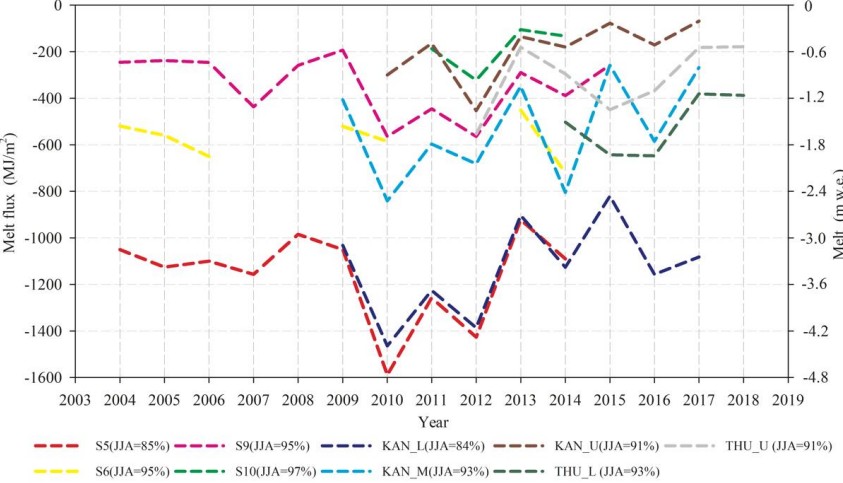


**Fig 7.** Annual melt energy (2003-2018) at the nine AWS sites and JJA melt energy percentage of
the annual total. Dashed line is the annual melt energy (MJ/m$^2$) and the right y-axis represents the
approximate melt water equivalent (m w.e. ).

Figure 8 presents the multi-year average seasonal cycle of 2 m temperature, 2 m
specific humidity and wind speed at 10 m at the nine AWS sites while Figure 9 shows
the multi-year average seasonal cycle of SEB components. Temperature and melt peak
in July for all sites. Average JJA $T_{2m}$ decreases with increasing latitude from 3.0 ℃ at
KAN_L to 1.4 ℃ at THU_L. The JJA elevational temperature gradient along the
K-transect is obvious with 3.7 ℃ at S5 decreasing to -3.0 ℃ at S10. Specific
humidity largely follows temperature. Wind speeds are katabatic in nature and
generally stronger in winter than in summer for the K-transect AWS sites. The
exception is S5 where wind speed shows a double peak because of persistent surface
melting in summer, i.e. like winter generating a situation with a colder surface and
warmer overlying air, generating persistent glacier winds. These higher wind speeds
enable the highest values for $Q_h$ for S5 as the strong wind shear enhances turbulent
mixing in summer, in spite of the strongly stable stratification (Figure 9). The average
summertime wind speeds at the T- transect AWS (7.2 m/s at THU_L and 6.6 m/s at



THU_U) are generally higher than at similar elevations along the K- transect (5.5 m/s
at KAN_M and 5.8 m/s at S10), and show a less well developed seasonal cycle,
possible owing to stronger synoptic forcing and higher cloud cover which limits
surface cooling to drive katabatic flow.

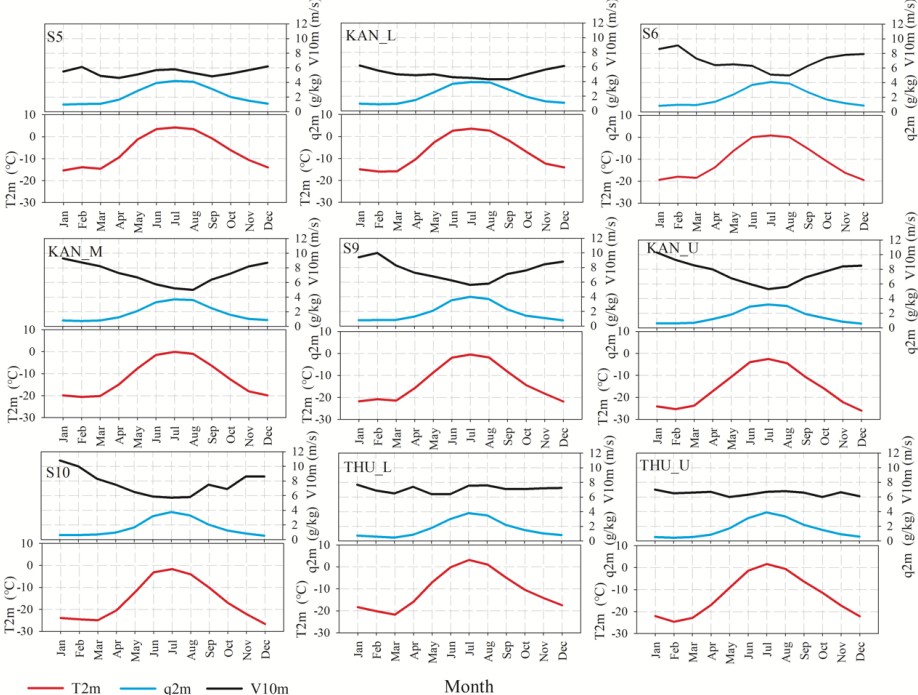


**Fig 8.** Multi-year average seasonal cycle based on monthly means of 2 m temperature (red, $T_{2m}$),
specific humidity (blue, $q_{2m}$) calculated from relative humidity and wind speed at 10 m (black,
$V_{10m}$).

Figure 9 shows the seasonal cycle of SEB components. M peaks in July at all
sites, mainly following $R_{net}$. But July melt differences with June are small at the lower
stations S5 and KAN_L where low wintertime accumulation means that the albedo
assumes the lower ice value early in the melt season, meaning that the main energy
source for melt, $S_{net}$, peaks at the end of June around the summer solstice. Melting
occurs as early as March and lasts until September at S5 and KAN_L, while S6 and
KAN_M also experience some melting in September. At THU_L and THU_U the
sharp peak in $S_{net}$ illustrates the shorter summer melt period.
For the lower AWS sites (S5, KAN_L, S6, KAN_M and THU_L), the shape of
the $L_{net}$ curve is relatively flat or even shows a maximum in summer. This is again a
signature of persistent surface melt at these lower sites, with the surface temperature
limited to a constant 273.15 K, limiting longwave heat loss from the surface
irrespective of $L_{in}$ (*Van den Broeke, et al., 2011*). For the higher AWS sites (S9,



KAN_U, S10 and THU_U) a minimum is reached later in spring, because the surface
is not yet melting and can still increase its temperature (and therewith $L_{out}$) in
response to increased absorption of solar radiation ($S_{net}$), at least for part of the day.

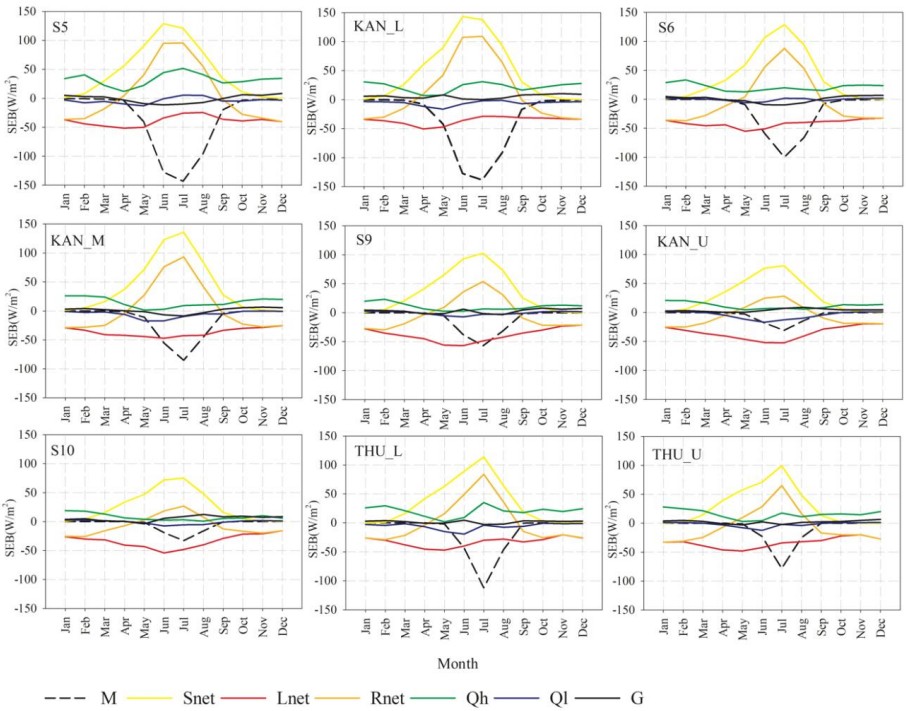


**Fig 9.**  Multi-year average seasonal cycle based on monthly means of SEB components.

The shapes of the seasonal $Q_h$ cycle at different AWS sites differ significantly.
All stations show a maximum in winter, reflecting that $Q_h$ is the most efficient SEB
component to balance $L_{net}$; the turbulent cooling of the air over the sloping ice sheet
surface results in katabatic winds that effectively mix the near surface air. In summer,
a second maximum occurs at S5, KAN_L and THU_L. These low-lying stations are
reached by relatively warm air in summer as shown in Figure 8, creating a strong
temperature gradient with the melting ice sheet, resulting in shallow katabatic flow
(glacier winds) and hence a large $Q_h$ that contributes significantly to melt (*Van den
Broeke, 1996; Van den Broeke et al., 2005*). At S5, KAN_L and THU_L, JJA $Q_h$
averages 45, 28, and 21 W m$^{-2}$, respectively, at least double that of the more elevated
and hence colder inland sites (KAN_M: 8 W m$^{-2}$ and KAN_U: 7 W m$^{-2}$). The latent
heat flux is generally small and negative, again with the exception of the lowest
stations where the persistent melting limits saturation specific humidity at the surface,
enabling condensation, making $Q_l$ a small heat source for melting. The strongest
sublimation rates are found in spring at the higher stations, when the sun heats the
surface without it reaching the melting point, enhancing the moisture gradient from
the surface to the near-surface air. Seasonal changes in G are small in comparison





with the other SEB components.
4.1.3 Variations of surface energy flux with elevation (K-transect)
The seven AWS along the K-transect enable the construction of robust JJA
SEB-elevation profiles (Fig 10). The average albedo in JJA (June, July and August),
calculated by dividing the total cumulative JJA values of $S_{out}$ and $S_{in}$, of S5, KAN_L
and S6 all were under 0.6, at KAN_M and S9 values were between 0.6 ~ 0.7, and at
KAN_U and S10 all values were higher than 0.7. Figure 10 shows that the magnitude
of the melt energy M decreases significantly as the elevation increases, from 122 W
$m^{-2}$ at S5 to 20 W $m^{-2}$ at S10, in line with $S_{net}$ which changes from 125 W $m^{-2}$ to 65
W $m^{-2}$ and $Q_h$ which decreases from 45 to 3 W $m^{-2}$, merely reflecting lower air
temperatures and a shorter melt season at the inland sites. $Q_l$ decreases from near zero
to being significantly negative (-12 W $m^{-2}$) at S10, reflecting significant surface
cooling by sublimation. Net longwave radiation also becomes a more dominant
surface heat sink at higher elevations. These profiles are valuable for the evaluation of
reanalysis products and (regional) climate models that are used to simulate and predict
melting at the surface of the GrIS. For several climate products this is done in the next
section.

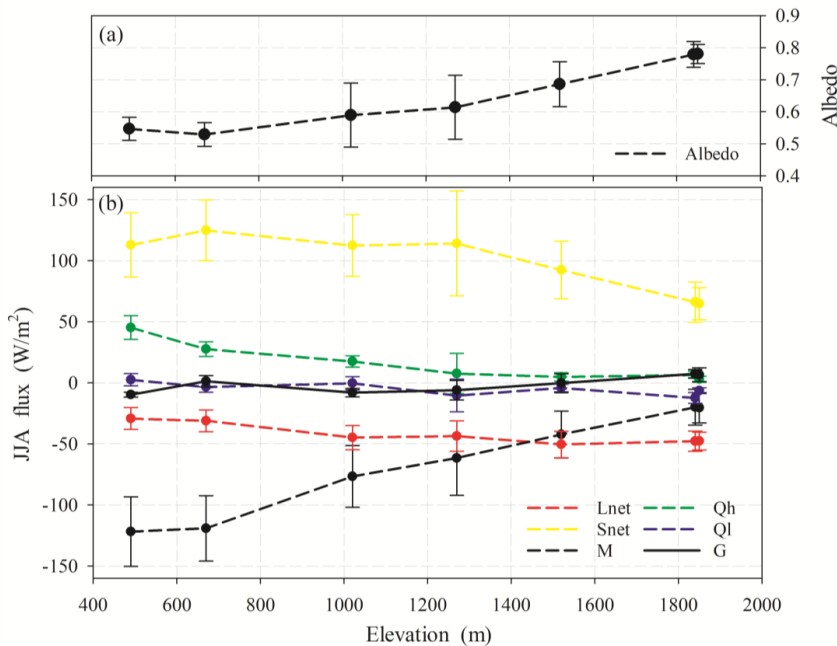


**Fig 10.**    Mean June, July, August (JJA) SEB components and albedo versus elevation along the
K-transect. Error bars indicate standard deviation in the multi-year annual mean.

**4.2 SEB evaluation in ERA5, ERA-Interim and RACMO2.3**





We use the results presented in the previous section to evaluate T$_{2m}$, albedo,
radiation fluxes, Q$_h$ and Q$_l$ in ERA5, ERA-interim, and RACMO2.3p2, the latter
forced at the lateral boundaries by ERA-interim during 2003-2018. We compute
model output at the AWS locations using an average distance-weighted interpolation
method using the four nearest grid points. Evaluation of KAN_L, KAN_M, KAN_U,
THU_L and THU_U are included in the Supplementary Materials, and the evaluation
of S5, S6, S9 and S10 can be found in *Noël et al., (2018)*. Tables S2-S6 (In the
Supplementary Materials) show the root mean square error (RMSE), the mean bias
(MB) and the correlation coefficient (R) based on linear regressions on daily
observations of the PROMICE AWS. KAN_L observations of T$_{2m}$ are well
represented in each of the models, but the differences with SEB components are
significant. Although ERA-5 better represents the observations than ERA-interim, the
improvement is not statistically significant (*Delhasse et al., 2019*). For Q$_h$ and Q$_l$,
RACMO2.3 provides the highest correlations. For THU_U (Table S3), RACMO2.3
shows high correlation coefficients for shortwave fluxes and 2 m temperature, and Q$_h$
and Q$_l$ are also relatively well represented with correlation coefficients between 0.8
and 0.7, higher than the ERA-reanalyses. For albedo, ERA5 performs significantly
better than ERA-interim both at KAN_L and THU_U. It turns out that ERA5 does not
significantly improve on ERA-Interim for near-surface climate and SEB in west
Greenland, apart from albedo. RACMO2.3 outperforms the reanalyses both for 2 m
temperature and SEB components, with results similar to *Noël et al., (2018)*.

### 4.3 Discussion

To better understand the processes driving intra-seasonal and inter-annual SEB
variability in west Greenland, we combine the SEB results presented above with
indices of two dominant regional circulation patterns: the Greenland Blocking Index
(GBI, *Hanna et al., 2015*) and the North Atlantic Oscillation index (NAO, *Hurrell et
al., 1995; Jones et al., 2003*).
Figure 11 presents the linear regression slope values of NAO and GBI with
monthly mean AWS JJA SEB components and 2 m temperatures, with units W m$^{-2}$ or
K per one standard deviation change in GBI ($\sigma_{GBI}$) and NAO ($\sigma_{NAO}$). The error bars
indicate the uncertainty in the regression slope, which generally shows stations along
the T-transect having a higher uncertainty than along the K-transect, mainly caused by
the shorter time series in combination with large interannual variability. The
associated Pearson correlation coefficients (R) are presented in the Supplementary
Materials. For instance, Figure S1 shows that significant positive correlations between
JJA AWS melt fluxes, T$_{2m}$ and the GBI are found for all AWS, whereas correlations
with NAO are weaker and generally negative (Figure S1a, b). For individual SEB
components S$_{net}$, L$_{net}$, Q$_h$ and Q$_l$, correlations reach significance for some but not all
stations, but again are generally stronger for GBI than for NAO (Figure. S1 c-f).
In Figure 11 several interesting features can be identified. Starting with GBI (red
symbols), we find significantly positive dependencies between JJA AWS melt fluxes





and GBI for all AWS (Fig. 11a). Along the K-transect, the dependency decreases from
a maximum of 13 W m$^{-2}$/$\sigma_{GBI}$ at S5 to ~5 W m$^{-2}$/$\sigma_{GBI}$ at S10 and KAN_U. The
dependencies of the individual SEB components along the K-transect are such that the
increase in $S_{net}$ (Fig. 11c) explains most (40-100%) of this melt increase, indicative of
clearsky conditions during episodes of large positive GBI, in agreement with previous
work (Hofer and others, 2018). Smaller contributions to the melt energy are made by
$Q_h$ (Fig. 11e) and $Q_l$ (Fig. 11f), the latter becoming significant because of the limiting
effect of surface melt on the surface temperature and hence its (saturated) specific
humidity, decreasing the sublimation potential (i.e. making $Q_l$ less negative). $L_{net}$ (Fig.
11d) contributes positively for the low-lying stations, again owing to the maximized
surface temperature during melt, limiting $L_{out}$, and negatively for the higher stations, a
result of enhanced surface cooling under clearsky, non-melting conditions. Surface
melt also modulates the 2 m temperature response (Fig. 11b), with a muted response
for the lower stations where melt is semi-permanent, and larger values at the higher
stations, where melt is intermittent.
Albeit with larger uncertainties, consistently high melt sensitivities to variations
in GBI of >15 W m$^{-2}$/$\sigma_{GBI}$ are found at THU_L and THU_U. Also here, the largest
contribution is made by $S_{net}$, but we find significant and approximately equal
contributions from $L_{net}$, $Q_h$ and $Q_l$. This suggests that in the northwest, high melt
under high GBI conditions is associated with high temperatures and cloudiness.

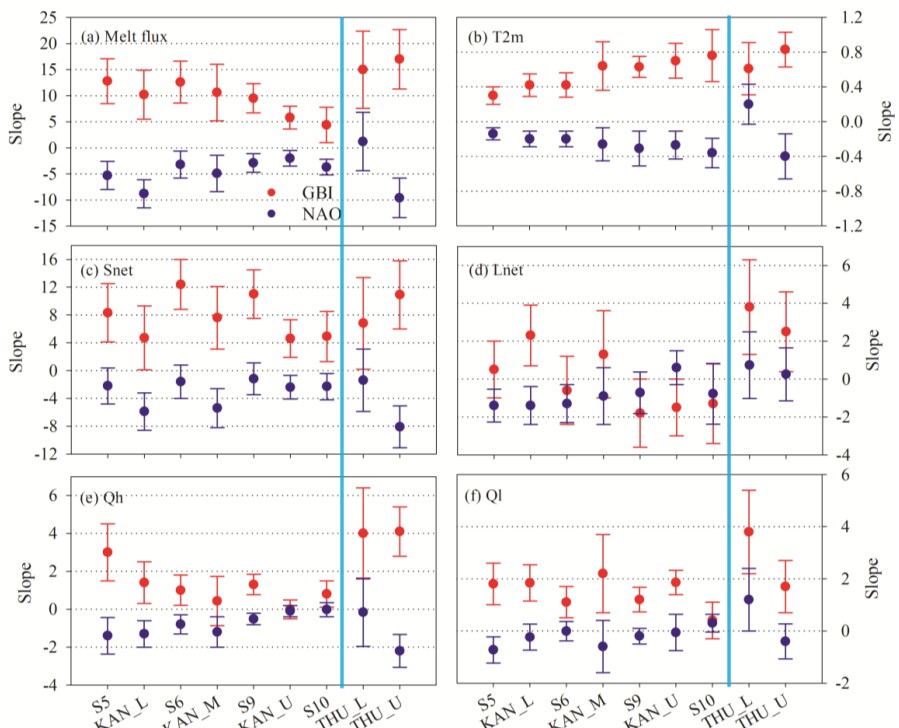




**Fig 11.** AWS regression slope of JJA average SEB components and 2 m temperature (T2m) with GBI (red dots) and NAO index (blue dots). Y axis are scaled with one standard deviation change in GBI/NAO circulation index to show (a) the melt flux change from SEB model in W $m^{-2}/\sigma_{GBI,NAO}$, (b) 2m temperature change from station in Kelvin$/\sigma_{GBI,NAO}$, (c) $S_{net}$ change from station in W $m^{-2}/\sigma_{GBI,NAO}$, (d) $L_{net}$ change from station in W $m^{-2}/\sigma_{GBI,NAO}$, (e) $Q_h$ change from SEB model in W $m^{-2}/\sigma_{GBI,NAO}$ and (f) $Q_l$ change from SEB model in W $m^{-2}/\sigma_{GBI,NAO}$. Error bars indicate standard error in the multi-year JJA mean.

To better understand this spatially different response of western GrIS climate and melt to GBI, Fig. 12 shows maps of the JJA GBI dependency for temperature (Fig.12a) and melt (Fig. 12c) for Greenland and its immediate surroundings from RACMO2, and Fig. 13a shows the regional 500 hPa height anomaly from ERA5. In the latter figure we use ERA5 since the RACMO2 domain does not cover the whole of the Arctic region. Both Figures 12 and 13 are based on data for the period 2000-2018 (19 years, 57 summer months). Figure S2 in the Supplementary Materials shows the correlation coefficient of 2 m temperature and melt flux of RACMO2.3 with the JJA GBI (Fig. S2a, S2c). Figure S2a shows R values for JJA 2 m temperature and GBI of 0.4-0.6 over the southwestern GrIS, very similar to the AWS results.

Figs. 12a, c confirm that the 2 m temperature/melt response to GBI are dominant in west Greenland and weaker towards the east. The maps also confirm the observed increasing/decreasing temperature/melt response with elevation in Figs. 11a, b under high GBI conditions along the K-transect in the southwestern GrIS, and the enhanced sensitivity in the northwest (Figs.12e and f show the enlarged images for melt). Fig. 13 shows that the large-scale circulation anomalies for high GBI conditions are very different for the southwestern and northwestern GrIS: the maximum positive anomaly is centered over the K-transect in the southwest, with the largest correlation coefficient R (Fig. S3 in the Supplementary Materials) causing clearsky conditions and a weak or absent circulation anomaly, which explains the dominant contribution of $S_{net}$ to the melt energy (*Hofer et al., 2017*). In the northwest, a significant circulation anomaly from the south and west means advection of warm and humid air, resulting in higher temperatures and enhanced cloudiness, which explains the more important contributions made to the melt anomaly by $L_{net}$, $Q_h$ and $Q_l$ (*Noël et al., 2019*).

Since 2007, the GBI has been predominantly positive in summer (Figure 3), with the exception of low-melt summers 2013 and 2018, and the strongest positive anomalies in the strong melt summers 2012 and 2015 (*Hanna et al., 2016*). High summer GBI episodes are clearly linked to exceptional GrIS melt years (*Hanna et al., 2014*), but *Hanna et al., (2013)* our results highlight the complexity of the response to summer GBI. *Young-Kwon Lim et al (2016)* show that in general, high pressure blocking primarily impacts the western areas of the GrIS via advective temperature increases. *Rimbu and Lohmann., (2011)* also found strong correlations between winter temperatures across southwestern GrIS and high blocking activity in the GrIS, whereas *Hanna et al., (2013)* show that temperatures in Tasiilaq (southeast Greenland)





do not show significant correlations with GBI. Here we confirmed and discussed these
different responses.
Dependencies of summer AWS melt and 2 m temperatures with NAO are
negative and generally weaker (Fig. 11, blue dots), implying a weaker influence of the
NAO on western GrIS near-surface climate and melt compared to the GBI. Fig. 13b
confirms a weaker and less organized impact of NAO on the large-scale circulation in
west Greenland, with two centres of action in the area of the Icelandic Low in
southeast Greenland, and a secondary centre over the Arctic. *Hanna et al., (2015)*
noted that the more local geographic nature of the GBI means that it correlates more
directly with Greenland climate than the NAO index, and our results support this.
Several studies identified a link between anomalously high air temperatures over the
GrIS during negative NAO phases (*Hanna and Cappelen, 2003; Chylek et al., 2004*).
A negative NAO index (high air surface pressures in the North Atlantic) is often
accompanied by anticyclonic ridging in the GrIS region (*Rajewicz et al., 2014*). Our
results suggest that both GBI and NAO affect the southern GrIS, this part of the ice
sheet being wetter during NAO positive phases, while drier when GBI is positive.
*Davini et al (2012)* noted that the geographical dependence of GrIS climate on the
NAO shifted eastward, which is consistent with an increase in GBI. Given the large
natural, interannual variability, it remains difficult at present to exactly partition the
contributions of atmospheric circulation variability and Arctic warming to intensive
melting in the western GrIS. Our regression analysis may further help to explain the
melting pattern of the western GrIS from the perspective of circulation anomalies
(*Hanna and Cappelen 2003; Overland and Wang, 2010; Overland et al., 2012*). Also
note in Fig.12 how Svalbard temperature and melt show opposite responses to GBI
compared to west Greenland (*Young-Kwon Lim et al., 2016*).

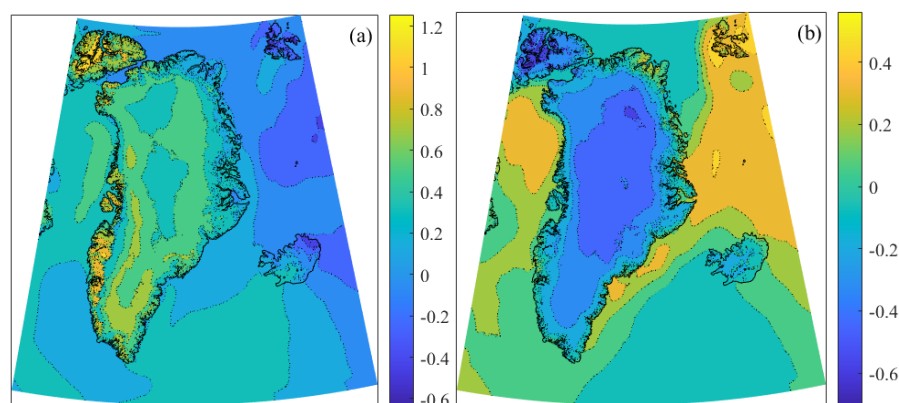



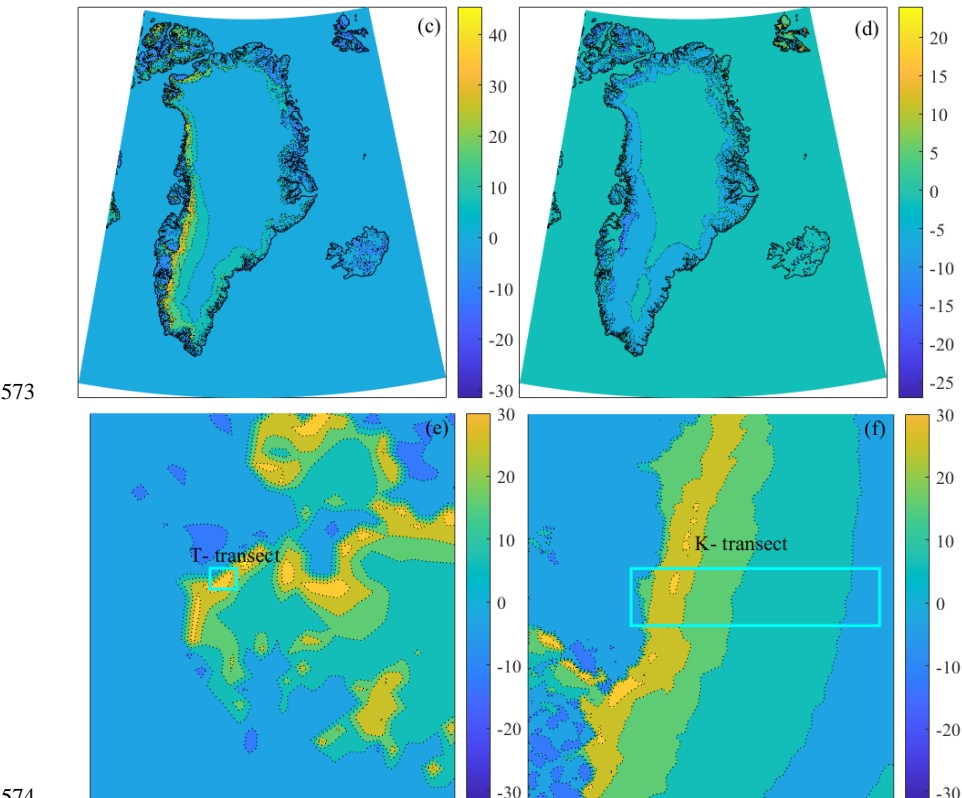



Fig 12. Regression slope of 2000~2018 JJA average 2 m temperature (T2m) from RACMO2.3
with (a) GBI and (b) NAO, melt flux from RACMO2.3 with (c) GBI and (d) NAO index.
Regression slope maps are scaled to show the 2m temperature change from RACMO2.3 in Kelvin
and melt flux change in W m$^{-2}$ for a one standard deviation change in GBI/NAO circulation index.
(e) and (f) are enlarged slope value image for T-transect and K-transect of JJA average melt flux
from RACMO2.3 with GBI. Black solid lines are Land-Sea Mask.

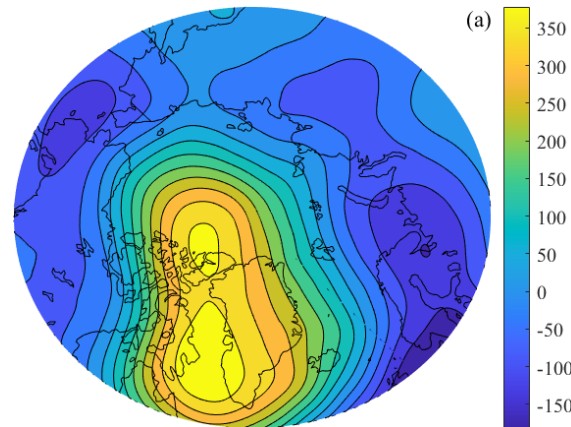

581
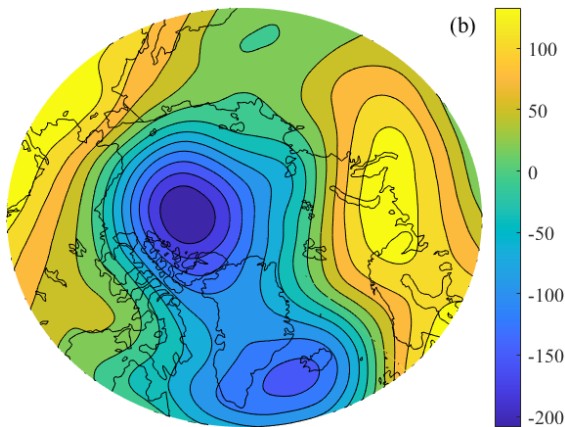

582

**Fig 13.** Regression fields slope of 2000~2018 JJA 500hpa geopotential height from ERA5 regressed with GBI (a) and NAO (b) index. Slope maps are scaled to show the 500hpa geopotential height change from ERA5 in geopotential metres change in gpm for a one standard deviation change in GBI (a) and NAO (b) circulation index.

## 5   Summary and conclusions

In this study, we forced a surface energy balance (SEB) model with data from nine automatic weather stations (AWS) situated in the southwestern (seven) and northwestern (two) Greenland ice sheet (GrIS). Absorbed shortwave radiation ($S_{net}$) is the main energy source for melting (M), followed by the sensible heat flux ($Q_h$). The multi-year average seasonal cycle of SEB components show that $S_{net}$ and M all peak in July, but that June is almost a similar strong melt month for the lowest stations. As the length of the melt season and average albedo in JJA decrease with elevation, so does melt; stations below 1,000 m asl show albedo values < 0.6, while the higher stations have > 0.7. $Q_h$ and the latent heat flux ($Q_l$) also decrease significantly with elevation, and the latter becomes negative at higher elevations, partly offsetting $Q_h$ as a surface heat source.

We used the AWS-derived near-surface climate variables and SEB components to evaluate the performance of two ECMWF reanalysis products (ERA5 and ERA-Interim) as a regional climate model RACMO2.3. Only for albedo does the newer ERA5 product significantly improve on ERA-Interim. The regional climate model RACMO2.3 has higher resolution (5.5 km) and a dedicated snow/ice module, and unsurprisingly outperforms the re-analyses.

From the decade-long observational time series, we inferred significant inter-annual variability in melt energy and SEB components, hiding any significant long-term trend. We report a strong positive correlation of the Greenland Blocking Index (GBI) with western GrIS melt and 2m temperature, and weaker and negative correlations with time series of summertime North Atlantic Oscillation (NAO) index.



## Supplementary Materials

The following supporting information is available as part of this article:

Figure S1. AWS correlations of JJA average SEB components and 2 m temperature (T2m) with GBI (red dots) and NAO index (blue dots).

Figure S2. Correlation fields of 2000~2018 JJA average 2 m temperature (T2m) from RACMO2.3 with (a) GBI and (b) NAO, melt flux from RACMO2.3 with (c) GBI and (d) NAO index.

Figure S3. Regression fields of 2000~2018 JJA 500hpa geopotential height regressed with GBI (a) and NAO (b) index. The color bars show the correlation coefficient R.

Table S1   Annual surface energy fluxes (W m$^{-2}$) at the nine AWS locations, SEB values of $L_{out}$, $Q_h$, $Q_l$, G and M are derived from the SEB model while $S_{in}$, $S_{out}$ and $L_{in}$ are from observations.

Table S2   Root Mean Squared Error (RMSE), mean bias (MB) and correlation coefficient (R) between daily AWS observations and ERA-Interim (EI), ERA5 (E5), RACMO2.3 (RAC) at KAN_L

Table S3   Root Mean Squared Error (RMSE), mean bias (MB) and correlation coefficient (R) between daily AWS observations and ERA-Interim (EI), ERA5 (E5), RACMO2.3 (RAC) at KAN_M

Table S4   Root Mean Squared Error (RMSE), mean bias (MB) and correlation coefficient (R) between daily AWS observations and ERA-Interim (EI), ERA5 (E5), RACMO2.3 (RAC) at KAN_U

Table S5   Root Mean Squared Error (RMSE), mean bias (MB) and correlation coefficient (R) between daily AWS observations and ERA-Interim (EI), ERA5 (E5), RACMO2.3 (RAC) at THU_L

Table S6   Root Mean Squared Error (RMSE), mean bias (MB) and correlation coefficient (R) between daily AWS observations and ERA-Interim (EI), ERA5 (E5), RACMO2.3 (RAC) at THU_U

**Data availability.** The micrometeorological observations are available from the Programme for Monitoring of the Greenland Ice Sheet (PROMICE) at http://promice.org/DataDownload.html, and the ERA-Interim and ERA5 re-analyses are available from the ECMWF at https://www.ecmwf.int/en/forecasts/datasets/reanalysis-datasets. All the results are available through an email request to the authors.

**Author contributions.** MRB provided the topic and idea, BJH, MRB and CHR coordinated the study and carried out the analysis; BJH and MRB drafted the paper, CHR edited the paper. All authors contributed to the analysis, discussion and interpretation of the results.

**Competing interests.** The authors declare that they have no conflict of interest.

## Acknowledgements



This work was funded by the Natural Science Foundation of China (41701059),
the Postdoctoral Science Foundation of China (40411594) and the Project for Outstanding Youth
Innovation Team in the Universities of Shandong Province (2019KJH011). The authors gratefully
acknowledge support from the Institute of Tibetan Plateau and Polar Meteorology, and data
availability from the Programme for Monitoring of the Greenland Ice Sheet (PROMICE) and the
ERA-Interim and ERA5 re-analyses projects of the ECMWF. The authors thank Brice Noël
(Utrecht University) for RACMO2 data support, Robert Fausto (GEUS) and Paul Smeets (Utrecht
University) for PROMICE and K-transect AWS technical information, and SEB model technical
support from Peter Kuipers Munneke, Maurice van Tiggelen and Constantijn Jakobs (all Utrecht
University).

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
