# Peer review of "Long-term surface energy balance of the western Greenland ice sheet"

_The Cryosphere, 2020_

## Referee Comment (RC1) · Anonymous Referee #1 · 4 Jul 2020

General comments

In this manuscript, the authors present the surface energy balance throughout western Greenland over a long-term AWS recording period. The paper provides a nice, detailed process-based analysis of similarities and differences in the surface energy balance between sites, and the controls exerted by elevation and latitude on these observed differences. Another strength of this work is that it provides one of the first validation studies of the ERA5 reanalysis over the Greenland Ice Sheet, finding that this dataset improves upon ERA-Interim only in the representation of albedo and is still inferior to the RACMO model. Although the scientific results presented in the paper are generally

timely and sound, it suffers from poor presentation quality in a number of instances, particularly related to the figures. In my opinion these issues, as detailed in the specific comments and technical corrections below, must be amended before I can recommend publication of this paper in The Cryosphere.

Specific comments

L11-12: What constitutes "reasonable agreement" between modeled and observed melt? Be more specific with numbers here, as you were with the RMSEs for modeled versus observed surface temperatures.

L138-139: This sentence requires a lot of work from the reader, forcing them to scroll up and down repeatedly to compare Figs. 2, 4, and 6. It would help to add short descriptions of each figure to the sentence, i.e. something like "Data points used in the model validation scatter plots (Figure 4) coincide with the time series of each variable in Figure 2, while the analysis of surface height changes in Figure 6 starts in 2008."

Figure 2: This figure would be easier to read with tighter y-axis limits, particularly the lower limits. Temperatures appear to never exceed ∼10C or drop below ∼-45C at any station, so the y-axis ranges can be made smaller.

Figure 2: This figure would also be more effective if the x-axes on all panels covered the same time period (i.e. ending all time series plots in 2019 instead of ending some in 2017 and others in 2019), even if the data at all stations is not available for the same record length. Why do the S5, S6, S9, and S10 records only extend through late 2016? The data gaps at S6 during 2008, 2010, 2012, and 2015 are addressed in the text, but I don't see anywhere in the text where the truncation of these records in 2016 is addressed.

L217-223: How are periods of ice- and snow-covered surface determined? Is this information derived from albedo calculations using the station data?

L259-260: It is again not entirely clear what "reasonable agreement" means in quantitative terms here. Is "reasonable agreement" a qualitative assessment of the percentage differences reported in the last sentence of the next paragraph?

L291-294: More recent references on the post-2012 slowdown in GrIS mass loss could be cited here and in L351-353, including Mouginot et al. (2019) and the Shepherd et al. (2019) paper already cited in the introduction.

Figure 6 and Figure 7: The color-coding of lines in these figures should be matched so that the same stations are plotted with the same colors in each figure, unless there is a compelling reason not to do this.

Figure 7: In my opinion, this figure would be easier to intuitively interpret if the y-axes were flipped so that more positive values indicate greater melt.

General comment on figures: Text size should be increased by a moderate amount on all figures to increase legibility. Text in Figures 8, 9, and 11 is particularly difficult to read at normal zoom levels for reading.

Figure 8: Why are q2m and 10-m wind speed plotted on a shared y-axis? I understand the authors need to conserve the use of space in this multi-panel figure, but it seems more logical that q2m would be plotted on the same panels as t2m, as these two variables are measured at the same level and can be expected to be highly correlated due to Clausius-Clapeyron.

L370-371: The meaning of the sentence "Specific humidity largely follows temperature" is not entirely clear. I assume it means that specific humidity increases alongside temperature due to the greater water vapor capacity of warmer air, but this should be stated clearly if so.

Figure 9: Following the sign conventions of the SEB terms on this plot, should melt energy values be positive?

L405-409: From examining Figure 9, it appears the summer Qh peak is as high or higher than the winter peak at S5, KAN_L, and THU_L. Consider rephrasing L406 to

say that "*Most* stations show a maximum in winter..."

L460-462: Can the authors provide any reasoning as to why ERA5 improves on ERA-Interim for albedo? Is there perhaps a revised surface scheme for ice sheets in ERA5?

L463: Why is this section called the "Discussion"? It seems to me to be a continuation of the reporting of the study results, and it doesn't seem that there is a separate "discussion" section in this paper per se. I would recommend renaming this section.

Figures 11-13: For clarity, units of the regression slope (W m-2 / $\sigma$ňGBI,NAO) should be included as labels of the y-axes in Figure 11 and the colorbars in Figures 12-13.

Figure 12 and Figure 13: The color ramps on these maps appear more suited for sequential rather than diverging data. To make it easier for the reader to discern areas of positive and negative regression slopes, a diverging color ramp should be used (for example the cmocean "balance" color map, for implementation in MATLAB see https://www.mathworks.com/matlabcentral/fileexchange/57773-cmocean-perceptually-uniform-colormaps). Additionally, the color scale should be set with a midpoint at zero to ensure that warm colors show above-zero values and cool colors show below-zero values.

Figure 12: Coastlines should be included in panels e and f.

L532-534: Figure 13 can be used to infer southwesterly advection of warm and humid air to NW Greenland given the spatial pattern of the regression slope. However, the maps do not show actual 500 hPa height contours or any wind vectors, so the figure does not directly show this phenomenon. This sentence should be amended to clearly indicate that this southwesterly advection has been inferred rather than directly shown.

Technical corrections

Title, L6-7, and elsewhere: My personal preference is to capitalize "Ice Sheet" in the phrase "Greenland Ice Sheet", since these letters are capitalized in the abbreviation "GrIS" throughout the paper. However I recognize that opinions differ on this and will

leave it to the authors' discretion.

L14: "show" –> "shows"

L19 and elsewhere: Capitalize the first "I" in "ERA-Interim"

L23 and elsewhere: Be consistent with using a hyphen or no hyphen in the words "reanalysis" and "reanalyes" (i.e. L23 says "re-analyses" while L24 says "reanalyses").

L52: "described" –> "have described"

L68: This is a misquote of Rajewicz and Marshall (2014). The actual quote from this paper is ". . . in Greenland" rather than ". . . in GrIS".

L77: Remove "i.e." from this sentence.

L83-85 (and elsewhere?): Be consistent with saying "over the GrIS" versus "over GrIS". L83 says "over GrIS" while L85 says "over the GrIS".

Table 2: There appear to be errors in text spacing and punctuation for the description of the wind speed and direction instruments. Are these sensors named "05103-5" and "05103-L" and manufactured by the same company (R. M. Young)?

Table 2: "Kipp Zonen" –> "Kipp & Zonen"

L149: "time" –> "temporal"

L164: This sentence would be easier to interpret with a comma after the word "evaluation".

L198: "sites" –> "site"

L318: "ice sheet" –> "ice sheet surface"

L488: "clearsky" –> "clear-sky"

L506: "axis" –> "axes"

L522: "response" –> "responses"

L539-542: This sentence doesn't make sense as written. Is the Hanna et al. 2013 reference supposed to be in the parentheses as an additional reference in support of the claim that "high summer GBI episodes are clearly linked to exceptional GrIS melt years"?

L575, 583: "∼" –> "-"

L592: "show" –> "shows"

L593: "similar" –> "similarly"

L601: "as" –> "and"

References

Mouginot, J., Rignot, E., Bjørk, A. A., van den Broeke, M., Millan, R., Morlighem, M., et al. (2019). Forty-six years of Greenland Ice Sheet mass balance from 1972 to 2018. Proceedings of the National Academy of Sciences, 116(19), 9239–9244. https://doi.org/10.1073/pnas.1904242116

---

## Referee Comment (RC2) · Federico Covi (Referee) · 7 Jul 2020

**General comments**

In this paper, Huai and coauthors use an energy balance model forced with automatic weather station data to compute surface energy fluxes along two transects in the west Greenland Ice Sheet. A detailed comparison and analysis of the surface energy fluxes is presented, with focus on differences due to elevation and latitude (e.g. different transect). Furthermore the differences between the two transects are put into a broader context using reanalysis products (ERA-Interim and ERA5) and a regional climate model (RACMO). The connection between Greenland Blocking Index

(GBI) and North Atlantic Oscillation (NAO) and the 2 m air temperature and melt flux is discussed. Finally a validation of the reanalysis products and RACMO is given using the results from the in-situ energy balance modeling for both near-surface climate variables and surface energy fluxes. While there are previous studies addressing the surface energy balance along the K-transect (e.g. Van den Broeke et al., (2008) and Kuipers Munneke et al., (2018), using data from the same AWS as in this study), the present work provides an unique contribution to the scientific community with its spatial analysis including the T-transect and the GBI/NAO indexes discussion. The manuscript is generally well organized and adequately presented. I would like to make two main comments/suggestions to be considered prior to publication in The Cryosphere:

1. **Section 4.2:** SEB evaluation in ERA5, ERA-Interim and RACMO2.3: this is a unique feature of the paper, as already commented by another reviewer it is "one of the first validation studies of the ERA5 reanalysis over the Greenland Ice Sheet". Yet only a half a page paragraph is reserved for the presentation/discussion of this evaluation, with all the tables shown in the Supplementary Material. While I do understand that this is not the actual primary scope of the paper, I think that the manuscript and the readers would benefit from a more exhaustive discussion of the evaluation, maybe including a summary figure/table in the main text. Perhaps some of the lengthy descriptions of the surface energy fluxes (Section 4.1.2) could be shortened. Similar results and discussions can be already found in previous works (e.g. Kuipers Munneke et al., (2018)) while the evaluation of ERA5, ERA-Interim and RACMO surface energy fluxes is a novelty. Please don't misunderstand me here, I am not suggesting to completely change the scope of the paper or its structure but just maybe to revise the balance of the results section.

2. **Validation of RACMO melt flux:** this is probably the main problem the paper has in its current version. Nowhere in the manuscript I have found a validation of RACMO modeled melt flux, yet this variable is used extensively in the discussion

section. In line 449 the reader is referred to Noël et al., (2018), whose SEB evaluation at sites S5, S6, S9, and S10 includes the melt flux as well (e.g. Tables 2-5). Ultimately modeling melt accurately is one of the final goals of any surface energy balance model and regional climate model used to study ice sheets changes. An often proposed and used approach to this type of validation is to evaluate point studies (e.g. like the SEB modeling results here presented) against in-situ observations and then evaluate regional climate models against the point studies. The current manuscript already uses this framework for all the SEB components, but in my opinion it needs to include the same analysis for the melt flux as well.

**Specific comments**

L52-58: I think there are other studies in the literature worth mentioning that specifically address the surface energy balance on the Greenland Ice Sheet (e.g. Charalampidis et al., (2015), Vandecrux et al., (2018)).

L76-88: the goals of the paper are not clearly stated before the structure is outlined. This paragraph could be streamlined to make the actual goals more straightforward. E.g.: we study the SEB at two transects . . . we put these results into a broader context using these products . . . which are validated in this way . . .

Table 1: I wonder if here ELA is used instead of elevation, also are some of the weather stations discontinued now? Would it be possible to put the full operational periods? (e.g. Start Date - End Date).

Section 2.2.1: are the AWS data processed? E.g. is any correction applied to the datasets before being used as model input? If yes processing procedures should be described here or referenced appropriately.

L121: to my understanding "emitted longwave radiation" is not used to drive the model but in the model evaluation.

L123-124: "where temperature is recalculated to the reference height of 2 m using

the SEB model" this procedure should be better explained or an appropriate reference given, if it's important.

L127-133: what is the surface height measurement used for? For model evaluation, as stated below, but is it also used as precipitation input for the model? It would be good to state what this data is used for and then describe its limitations and corrections applied.

L134-139: the phrasing of this paragraph could be improved to better deliver the message.

Figure 2: is this figure really needed just to show the data availability period?

Section 2.2.2 and 2.2.3: maybe these two sections could be merged to improve the readability of the paper. Also the title should include the ERA-Interim product.

L185-186: convoluted sentence, readability could be improved.

L192: what does the author mean exactly with the surface value of the calculated subsurface heat flux?

L215-217: the process of Smeets and van den Broeke., (2008) could be better explained or simply skipped (e.g. … following the study of Smeets and van den Broeke., (2008) a value of …).

L226-233: how is the subsurface part of the model initialized?

Figure 4: any comment on the fact that it appears that modeled surface temperature is 0C much more often than the observed, this could mean that modeled melt is overestimated.

L251-252: I find this sentence a bit misleading, there is still information to be retrieved from 0C surface temperature (e.g. see previous comment about Figure 4), however it is true that the amount of melt cannot be assessed just by using melting surface temperature as a proxy.

Figure 5: modeled data and observed data axis are inverted compared to Figure 4, this should be avoided at all cost. Be consistent with the chosen convention (e.g. modeled data always on the x-axis).

L275-279: such generalization should be avoided in my opinion when site characteristics vary so much. E.g. at S5 the radiation penetration effect on total cumulative melt flux is neglectable but what about at a much higher elevation site like S10 where the melt flux is much smaller? Section 4.1.1 and Figure 6: why there are no model values of surface height change for KANU and S10? Also doesn't the model simulate accumulation? Why are measured height changes compared to modelled ice melt in Figure 6? Also some of the dashed lines are not continuous (e.g. S6, KANL, . . .) why is this the case? A better explanation should be given here. (The nature of this comment is similar to my general comment about RACMO melt flux validation)

L309-311: reference about the cloud cover product used here?

Figure 8: I would rather put T2m and q2m on the same subplot (but different axis) since they are correlated rather than q2m with the wind.

Figure 11: this figure needs a bit of work from the reader to be fully understood. Providing additional descriptive text (e.g. at L470) to assist the reader would help. Also consider keeping the y-axis symmetric and with the same range. This would help in assessing the difference between different fluxes.

L512-515: convoluted sentence, readability could be improved.

Figure 12 and 13: missing subplot titles and colorbar labels.

**References**

Charalampidis, C., van As, D., Box, J. E., van den Broeke, M. R., Colgan, L. T., Doyle, S. H., et al. (2015). Changing surface-atmosphere energy exchange and refreezing capacity of the lower accumulation area, West Greenland. The Cryosphere, 9(6), 2163–2181.

Kuipers Munneke, P. C. J. P. P. Smeets, C. H. Reijmer, J. Oerlemans, R. S. W. van de Wal and M. R. van den Broeke.: The K-transect on the western Greenland Ice Sheet: Surface energy balance (2003-2016), Arct. Antarct. Alp., 50:1, e1420952, 2018.

Noël, B., van de Berg, W. J., van Meijgaard, E., van de Wal, R. S. W., and van den Broeke, M. R.: Modelling the climate and surface mass balance of polar ice sheets using RACMO2-Part1: Greenland (1958-2016), The Cryosphere, 12, 811-831, 2018.

Van den Broeke, M., Smeets, P., Ettema, J., van der Veen, C., van de Wal, R., and Oerlemans, J.: Partitioning of melt energy and meltwater fluxes in the ablation zone of the west Greenland ice sheet, The Cryosphere, 2, 179-189, 2008.

Vandecrux, B., Fausto, R. S., Langen, P. L., van As, D., MacFerrin, M., Colgan, W. T., et al. (2018). Drivers of firn density on the Greenland ice sheet revealed by weather station observations and modeling. Journal of Geophysical Research: Earth Surface, 123, 2563–2576.

---

## Author Comment (AC1) · 1 Aug 2020

**General comments**

In this manuscript, the authors present the surface energy balance throughout western Greenland over a long-term AWS recording period. The paper provides a nice, detailed process-based analysis of similarities and differences in the surface energy balance between sites, and the controls exerted by elevation and latitude on these observed differences. Another strength of this work is that it provides one of the first validation studies of the ERA5 reanalysis over the Greenland Ice Sheet, finding that this dataset improves upon ERA-Interim only in the representation of albedo and is still inferior to the RACMO model. Although the scientific results presented in the paper are generally timely and sound, it suffers from poor presentation quality in a number of instances, particularly related to the figures. In my opinion these issues, as detailed in the specific comments and technical corrections below, must be amended before I can recommend publication of this paper in The Cryosphere.

**Reply: We thank the reviewer for the comments, which have improved the paper.**

**Specific comments**

L11-12: What constitutes "reasonable agreement" between modeled and observed melt? Be more specific with numbers here, as you were with the RMSEs for modeled versus observed surface temperatures.

**Reply: We added specific numbers with explanation.**

The uncertainty in daily ablation measurements owing to different error sources (differential ablation, density of ice, stake reading) can be as large as ±10% (*Braithwaite et al., 1998*). *Van den Broeke et al., (2010)* report that constant systematic meteorological measurement errors, which can be interpreted as an upper bound on the modelled uncertainty range, result in model melt uncertainty of ±15%. Given these uncertainty estimates, with an average difference of 6% between observed and modelled ice melt, the agreement is reasonable.

L138-139: This sentence requires a lot of work from the reader, forcing them to scroll up and down repeatedly to compare Figs. 2, 4, and 6. It would help to add short descriptions of each figure to the sentence, i.e. something like "Data points used in the model validation scatter plots (Figure 4) coincide with the time series of each variable in Figure 2, while the analysis of surface height changes in Figure 6 starts in 2008."

**Reply: We followed the reviewer's advice and added:**

Data points used in the model validation scatter plots (Figure 4) coincide with the time series of 2 m temperature in Figure 2, while the analysis of surface height changes in Figure 6 starts in 2008.

Figure 2: This figure would be easier to read with tighter y-axis limits, particularly the lower limits. Temperatures appear to never exceed~10C or drop below~-45C at any station, so the y-axis ranges can be made smaller.

**Reply: The new Figure 2 y-axis range has been made smaller with the best range of -55 ℃~ 15 ℃.**

[Figure]

**New Figure 2**

Figure 2: This figure would also be more effective if the x-axes on all panels covered

the same time period (i.e. ending all time series plots in 2019 instead of ending some in 2017 and others in 2019), even if the data at all stations is not available for the same record length. Why do the S5, S6, S9, and S10 records only extend through late 2016? The data gaps at S6 during 2008, 2010, 2012, and 2015 are addressed in the text, but I don't see anywhere in the text where the truncation of these records in 2016 is addressed.

**Reply: Thank you for pointing this out. We have now added the used end date in Table 1. According to the comments, we fixed Figure 2 as requested:**

**(1)** We extend the data of S5, S6 and S9 stations to 2018, but S10 records only extend until 2016, as now indicated in Table 1.

**(2)** We also update the results of S5, S6 and S9 for the year of 2017, 2018 in all the figures and tables, and text.

**(3)** Regarding the data treatment and gaps, we give a detailed description in the third paragraph of "2.2.1 AWS data and processing". We indicate here that for annual or multi-year averages of SEB components, we only use complete years, therefore in the analysis of SEB, these gaps are not specifically addressed. Note that in Figures 6, 7 gaps occur for the same years, and we have added an explanation in the text.

L217-223: How are periods of ice- and snow-covered surface determined? Is this information derived from albedo calculations using the station data?

**Reply: Indeed, determining whether snow or ice is present at the surface is done by combining surface albedo and sonic height ranger data. This is now stated at Line 232-233.**

L259-260: It is again not entirely clear what "reasonable agreement" means in quantitative terms here. Is "reasonable agreement" a qualitative assessment of the percentage differences reported in the last sentence of the next paragraph?

**Reply: Please see reply to "L11-12" comments.**

L291-294: More recent references on the post-2012 slowdown in GrIS mass loss could be cited here and in L351-353, including Mouginot et al. (2019) and the Shepherd et al. (2019) paper already cited in the introduction.

**Reply: Here we have cited these two papers.**

The strongest melt occurred in summer 2012, contributing to the largest annual ice-sheet mass loss on record (*Khan et al., 2015; Mouginot et al., 2019; Shepherd et al., 2019*), followed by a return to more average conditions in 2013 (*Nghiem et al., 2012; Kuipers Munneke et al., 2018*).

Figure 6 and Figure 7: The color-coding of lines in these figures should be matched so that the same stations are plotted with the same colors in each figure, unless there is a

compelling reason not to do this.

**Reply: We re-plotted Figure 7 in order to make the line colors of Figure 6 and Figure 7 consistent.**

[Figure]

**New Figure 7**

Figure 7: In my opinion, this figure would be easier to intuitively interpret if the y-axes were flipped so that more positive values indicate greater melt.

**Reply: We prefer to retain the negative value for melt energy to keep the SEB equation consistent and the SEB figures clear. We add the explanation as follows:**

In the Surface Energy Balance (SEB) equation:

$$M+S_{in}+S_{out}+L_{in}+L_{out}+Q_h+Q_l+G+Q_p=0 \qquad (1)$$

all fluxes that are directed towards/away from the surface (surface gains/loses energy) are defined positive.

General comment on figures: Text size should be increased by a moderate amount on all figures to increase legibility. Text in Figures 8, 9, and 11 is particularly difficult to read at normal zoom levels for reading.

**Reply: We enlarged the font size in Figures 8, 9, and 11.**

Figure 8: Why are q2m and 10-m wind speed plotted on a shared y-axis? I understand the authors need to conserve the use of space in this multi-panel figure, but it seems more logical that q2m would be plotted on the same panels as t2m, as these two variables are measured at the same level and can be expected to be highly correlated due to Clausius-Clapeyron.

**Reply: Combining this suggestion with the suggestions from the second reviewer,**

**we separated the three variables. If q2m and t2m are plotted on the same subplot, it is difficult to distinguish the amplitude of the seasonal cycle of q2m.**

[Figure]

**New Figure 8**

L370-371: The meaning of the sentence "Specific humidity largely follows temperature" is not entirely clear. I assume it means that specific humidity increases alongside temperature due to the greater water vapor capacity of warmer air, but this should be stated clearly if so.

**Reply: We changed this sentence following the reviewer's suggestion.**

Specific humidity increases alongside temperature due to the greater water vapor capacity of warmer air, implying that specific humidity largely follows temperature.

Figure 9: Following the sign conventions of the SEB terms on this plot, should melt energy values be positive?

**Reply: Please see reply to "Figure 7" comments.**

L405-409: From examining Figure 9, it appears the summer Qh peak is as high or higher than the winter peak at S5, KAN_L, and THU_L. Consider rephrasing L406 to say that "*Most* stations show a maximum in winter..."

**Reply: Here we checked the value of Qh in summer, and then we fixed this sentence.**

Most stations show a maximum in winter, reflecting that $Q_h$ is the main SEB component to balance $L_{net}$.

L460-462: Can the authors provide any reasoning as to why ERA5 improves on ERA Interim for albedo? Is there perhaps a revised surface scheme for ice sheets in ERA5?

**Reply: we add the explanation as follow:**

This is probably caused by the new snow albedo scheme, which changes exponentially with snow age in ERA5, and resets fresh snow albedo, while ERA-Interim set a maximum constant albedo for snow events (*ECMWF, 2016*).

L463: Why is this section called the "Discussion"? It seems to me to be a continuation of the reporting of the study results, and it doesn't seem that there is a separate "discussion" section in this paper per se. I would recommend renaming this section.

**Reply: we re-named "4.3 Discussion" as "4.3 Relationships with large-scale circulation variability".**

Figures 11-13: For clarity, units of the regression slope (W m$^2$/σ GBI,NAO) should be included as labels of the y-axes in Figure 11 and the colorbars in Figures 12-13.

**Reply: we have added the slope units to Figure 11and Figures 12-13.**

[Figure]

**New figure 11**

Figure 12 and Figure 13: The color ramps on these maps appear more suited for sequential rather than diverging data. To make it easier for the reader to discern areas of positive and negative regression slopes, a diverging color ramp should be used (for example the cmocean "balance" color map, for implementation in MATLAB see https://www.mathworks.com/matlabcentral/fileexchange/57773cmocean-perceptually-uniform-colormaps). Additionally, the color scale should be set with a midpoint at zero to ensure that warm colors show above-zero values and cool colors show below-zero values.

**Reply: Thank you for the suggestion to use toolbox "CMOcean Colormaps".**

We now use "cmocean ('balance', 'pivot', 0)" for the new Figure 12 and Figure 13.

[Figure]

[Figure]

(e) T-transect M~GBI

(f) K-transect M~GBI

W m$^{-2}$/σ GBI

**New Figure 12**

[Figure]

(a) gpm/σ GBI

(b) gpm/σ NAO

**New Figure 13**

Figure 12: Coastlines should be included in panels e and f.

**Reply: We added coastlines to Figures 12 e and 12 f as shown above.**

L532-534: Figure 13 can be used to infer southwesterly advection of warm and humid air to NW Greenland given the spatial pattern of the regression slope. However, the maps do not show actual 500 hPa height contours or any wind vectors, so the figure does not directly show this phenomenon. This sentence should be amended to clearly indicate that this southwesterly advection has been inferred rather than directly shown.

**Reply:Here we changed the sentence as follows:**

Assuming geostrophy, the circulation anomalies in Fig. 13a imply anomalous southwesterly flow in northwest Greenland blocking conditions. Previous studies confirm that in the northwest, during blocking conditions anomalous southwesterly advection of warm and humid air results in higher temperatures and enhanced cloudiness, which explains the more important contributions made to the melt anomaly by $L_{net}$, $Q_h$ and $Q_l$ (Noël et al., 2019).

**Technical corrections**

Title, L6-7, and elsewhere: My personal preference is to capitalize "Ice Sheet" in the phrase "Greenland Ice Sheet", since these letters are capitalized in the abbreviation "GrIS" throughout the paper. However I recognize that opinions differ on this and will leave it to the authors' discretion.

**Reply:** We changed as "Greenland Ice Sheet".

L14: "show" –> "shows"

**Reply:Thank you, changed.**

The multi-year average seasonal cycle of SEB components shows that $S_{net}$ and M peak in July at all AWS.

L19 and elsewhere: Capitalize the first "I" in "ERA-Interim"

**Reply:Changed.**

L23 and elsewhere: Be consistent with using a hyphen or no hyphen in the words "reanalysis" and "reanalyes" (i.e. L23 says "re-analyses" while L24 says "reanalyses").

**Reply:Changed.**

L52: "described" –> "have described"

**Reply:Changed.**

L68: This is a misquote of Rajewicz and Marshall (2014). The actual quote from this

paper is "... in Greenland" rather than "... in GrIS".

**Reply:Changed.**

Table 2: There appear to be errors in text spacing and punctuation for the description of the wind speed and direction instruments. Are these sensors named "05103-5" and "05103-L" and manufactured by the same company (R. M. Young)?

**Reply:These sensors are from same company (R. M. Young05103).**

Table 2: "Kipp Zonen" –> "Kipp & Zonen"

**Reply:Changed.**

L149: "time" –> "temporal"

**Reply:Changed.**

L164: This sentence would be easier to interpret with a comma after the word "evaluation".

**Reply:Changed.**

L198: "sites" –> "site"

**Reply:Changed.**

L318: "ice sheet" –> "ice sheet surface"

**Reply:Changed.**

L488: "clearsky" –> "clear-sky"

**Reply:Changed.**

L506: "axis" –> "axes"

**Reply:Changed.**

L522: "response" –> "responses"

**Reply:Changed.**

L539-542: This sentence doesn't make sense as written. Is the Hanna et al. 2013 reference supposed to be in the parentheses as an additional reference in support of the claim that "high summer GBI episodes are clearly linked to exceptional GrIS melt years"?

**Reply:** We corrected this sentence to read:
High summer GBI episodes are clearly linked to exceptional GrIS melt years (*Hanna et al., 2013; 2014*), but *Hanna et al. (2013)* as well as our results highlight the complexity of the response to variations in summer GBI.

L575, 583: "~" –> "-"

**Reply:Changed.**

L592: "show" –> "shows"

**Reply:Changed.**

L593: "similar" –> "similarly"

**Reply:Changed.**

L601: "as" –> "and"

**Reply:Changed.**

**References**

**Reply:** We add these new references.

Mouginot, J., Rignot, E., Bjørk, A. A., van den Broeke, M., Millan, R., Morlighem, M., et al. (2019). Forty-six years of Greenland Ice Sheet mass balance from 1972 to 2018. Proceedings of the National Academy of Sciences, 116(19), 9239-9244. https://doi.org/10.1073/pnas.1904242116.

Braithwaite RJ, Konzelmann T, Marty C and Olesen OB (1998) Errors in daily ablation measurements in northern Greenland, 1993-94, and their implications for glacier climate studies. J. Glaciol., 44(148), 583-588.

Van den Broeke, M. R., König-Langlo, G., Picard, G., Kuipers Munneke, P., and Lenaerts, J. T. M.: Surface energy balance, melt and sublimation at Neumayer Station, East Antarctica, Antarct. Sci., 22, 87-96, https://doi.org/10.1017/S0954102009990538, 2010.

---

## Author Comment (AC2) · 1 Aug 2020

**We thank the reviewer for the comments, which have improved the paper.**

**General comments**

**1.   Section 4.2:**

SEB evaluation in ERA5, ERA-Interim and RACMO2.3: this is a unique feature of the paper, as already commented by another reviewer it is "one of the first validation studies of the ERA5 reanalysis over the Greenland Ice Sheet". Yet only a half a page paragraph is reserved for the presentation/discussion of this evaluation, with all the tables shown in the Supplementary Material. While I do understand that this is not the actual primary scope of the paper, I think that the manuscript and the readers would benefit from a more exhaustive discussion of the evaluation, maybe including a summary figure/table in the main text. Perhaps some of the lengthy descriptions of the surface energy fluxes (Section4.1.2) could be shortened. Similar results and discussions can be already found in previous works (e.g. Kuipers Munneke et al., (2018)) while the evaluation of ERA5, ERA-Interim and RACMO surface energy fluxes is a novelty. Please don't misunderstand me here, I am not suggesting to completely change the scope of the paper or its structure but just maybe to revise the balance of the results section.

**Reply: A recent ERA5 evaluation in Greenland was published in Delhasse et al., (2020), and we now make a reference and brief comparison with that study.**

We use the results presented in the previous section to evaluate $T_{2m}$, albedo, radiation fluxes, $Q_h$ and $Q_l$ in ERA5, ERA-Interim, and RACMO2.3p2, the latter forced at the lateral boundaries by ERA-Interim during 2003-2018. We compute model output at the AWS locations using an average distance-weighted interpolation method using the four nearest grid points. Evaluation of KAN_L, KAN_M, KAN_U, THU_L and THU_U are included in the Supplementary Materials, and the evaluation of S5, S6, S9 and S10 can be found in *Noël et al., (2018)*. Tables S2-S5 (In the Supplementary Materials) show the root mean square error (RMSE), the mean bias (MB) and the correlation coefficient (R) based on linear regressions on daily observations of the PROMICE AWS.

Although ERA5 better represents the observations than ERA-interim, the improvement is not statistically significant for all the near-surface variables, in agreement with *Delhasse et al., (2020)*. For $Q_h$ and $Q_l$, RACMO2.3 provides the highest correlations. For THU_U (Table S3), RACMO2.3 shows high correlation coefficients for shortwave fluxes and 2 m temperature, and $Q_h$ and $Q_l$ are also relatively well represented with correlation coefficients between 0.8 and 0.7, higher than both ERA reanalyses. For albedo, ERA5 outperforms ERA-Interim at most stations. This is likely caused by the new snow albedo scheme, which changes exponentially with snow age in ERA5, and resets fresh snow albedo, while ERA-Interim set a maximum constant albedo for snow events (*ECMWF, 2016*).

We conclude that the regional climate model RACMO2.3 remains a useful addition to reanalysis products for the simulation of GrIS near-surface climate and SEB.

**2.    Validation of RACMO melt flux:**

This is probably the main problem the paper has in its current version. Nowhere in the manuscript I have found a validation of RACMO modeled melt flux, yet this variable is used extensively in the discussion section. In line 449 the reader is referred to Noël et al., (2018),whose SEB evaluation at sites S5, S6, S9, and S10 includes the melt flux as well (e.g. Tables 2-5). Ultimately modeling melt accurately is one of the final goals of any surface energy balance model and regional climate model used to study ice sheets changes. An often proposed and used approach to this type of validation is to evaluate point studies (e.g. like the SEB modeling results here presented) against in-situ observations and then evaluate regional climate models against the point studies. The current manuscript already uses this framework for all the SEB components, but in my opinion it needs to include the same analysis for the melt flux as well.

**Reply:**

**(1)** A validation of SEB model against in-situ ablation rate observations is presented in Figure 5 (Average 10-day modeled and observed ice melt). This only works for ice with known density, i.e. at KAN_L, KAN_M, S5, S6 and THU_L which are situated below the equilibrium line. See for the detailed descriptions Section 3.2 (SEB model evaluation) and discussion of Figure 5.

**(2)** We added a row in the evaluation tables in which RACMO2.3p2 melt rate is compared to that from the SEB model for station KAN_L, KAN_M, KAN_U, THU_L and THU_U. Previously, an extensive evaluation of RACO2.3p2 ablation rate with all available observations from Greenland S5, S6, S9 and S10 was performed by *Noël et al., (2018)* and also showed good agreement.

**Specific comments**

L52-58: I think there are other studies in the literature worth mentioning that specifically address the surface energy balance on the Greenland Ice Sheet (e.g. Charalampidis et al., (2015), Vandecrux et al., (2018)).

**Reply: We added the suggested literature in the "Introduction".**

*Charalampidis et al., (2015)* use a surface energy balance model forced by five years of K-transect AWS measurements to evaluate the seasonal and interannual SEB variability, in particular the exceptionally warm summers of 2010 and 2012. *Vandecrux et al., (2018)* present a simulation of near-surface firn density in the percolation zone, to quantiy the influence of climatic drivers such as snowfall and surface melt.

L76-88: the goals of the paper are not clearly stated before the structure is outlined. This paragraph could be streamlined to make the actual goals more straightforward. E.g.: we study the SEB at two transects ... we put these results into a broader context using these products ... which are validated in this way ...

**Reply: We re-organized this paragraph as follows:**

We study the dependency of west Greenland SEB and melt on large-scale circulation variability along two GrIS AWS transects, i.e. the southwestern Kangerlussuaq (K-) transect and the northwestern Thule (T-) transect. We put these regional results into a broader spatial context using reanalysis (ERA5, ERA-Interim) products and output of a regional atmospheric climate model (RACMO2.3). ERA5 is the latest reanalysis product from the European Centre for Medium-Range Weather Forecasts (*ECMWF; Dee et al., 2011; Hersbach and Dee, 2016*), and replaces ERA-Interim, considered to be the leading product over GrIS until now (*Albergel et al., 2018; Bromwich et al., 2016*). Because both the PROMICE and IMAU AWS are not assimilated in ERA5, these data can be used to assess its quality and that of regional climate models. Thus, we also include an evaluation of ERA5/RACMO2.3 SEB components over the western GrIS.

Table1: I wonder if here ELA is used instead of elevation, also are some of the weather stations discontinued now? Would it be possible to put the full operational periods? (e.g. Start Date - End Date).

**Reply: we changed ELA to elevation. For the data periods of every station used in this study, we put them in Figure 2.**

**Table 1** AWS location, elevation and start of observations

| Station | Latitude(N) | Longitude(W) | Elevation (m a.s.l) | Start Date | End Date |
|---------|-------------|--------------|---------------------|------------|----------|
| S5 | 67.08 | 50.10 | 490 | 27/08/2003 | 01/01/2019 |
| S6 | 67.07 | 49.38 | 1020 | 01/01/2003 | 01/01/2019 |
| S9 | 67.05 | 48.22 | 1520 | 26/08/2003 | 27/08/2019 |
| S10* | 67.00 | 47.02 | 1850 | 17/08/2010 | 13/09/2016 |
| KAN_L | 67.10 | 49.95 | 670 | 01/09/2008 | 18/02/2018 |
| KAN_M | 67.07 | 48.84 | 1270 | 02/09/2008 | 18/02/2018 |
| KAN_U | 67.00 | 47.03 | 1840 | 04/04/2009 | 19/08/2018 |
| THU_L | 76.40 | 68.27 | 570 | 09/08/2010 | 05/10/2018 |
| THU_U | 76.42 | 68.15 | 760 | 09/08/2010 | 06/09/2018 |

*S10 is currently stopped while other stations are still operational.

Section 2.2.1: are the AWS data processed? E.g. is any correction applied to the datasets before being used as model input? If yes processing procedures should be described here or referenced appropriately.

**Reply: For the data processing, we have cited references from the data providers IMAU (*Smeets et al., 2018*) and GEUS (*Fausto et al., 2012b; Van As et al., 2011*).**

Snow and ice height records cannot always be used directly to assess sensor height changes because of AWS design changes and/or settling of the structure. For PROMICE AWS, we use the results from a physically based method to remove air-pressure variability from the signal of the pressure transducer records (*Fausto et al., 2012b; Van As et al., 2011*). For details of S5, S6, S9 and S10 data biases, corrections, and data gap filling in case of sensor failure, we refer to *Smeets et al. (2018)*.

L121: to my understanding "emitted longwave radiation" is not used to drive the model but in the model evaluation.

**Reply: Thank you, we removed "...and emitted..." and added at the end of the sentence: "and emitted longwave radiation is used to evaluate the model performance."**

L123-124: "where temperature is recalculated to the reference height of 2 m using the SEB model" this procedure should be better explained or an appropriate reference given, if it's important.

**Reply: we reformulated as follows:**

The height of the temperature/humidity sensor continuously changes due to ablation and/or accumulation and settling of the station. In order to compare to model output at the 2 m reference height, AWS temperature and humidity are recalculated to this height using the flux-profile relations applied to the turbulent fluxes from the SEB model. To illustrate the data time series at the nine AWS, Figure 2 shows the full record of 2 m temperature.

L127-133: what is the surface height measurement used for? For model evaluation, as stated below, but is it also used as precipitation input for the model? It would be good to state what this data is used for and then describe its limitations and corrections applied.

**Reply: Information about the surface height is required for turbulent flux calculations, to identify surface type for albedo, to feed snow accumulation into the model and for correction of wind, temperature and humidity to standard heights. We added:**

The sonic height ranger provides changes in the surface height, which allows us to accurately determine snow depth, surface type (ice/snow) for albedo, sensor height required for turbulent flux calculations as well as for correction of temperature and humidity values to standard height.

L134-139: the phrasing of this paragraph could be improved to better deliver the

message.

**Reply: we rephrased as follows:**

Note that AWS time series have differing lengths and completeness. For model evaluation with surface temperature (Fig. 4) we used all available hourly values of emitted longwave radiation, i.e. data points used for Figure 4 coincide with the time series as shown in Figure 2. The evaluation using observed ice melt (Fig. 6) uses data starting in 2008, to maximize overlap between the various AWS time series. For the calculation of the average SEB seasonal cycle we used only complete years (Tables 3 and S1, Figures 7, 8, 9 and 10).

Figure 2: is this figure really needed just to show the data availability period?

**Reply: We think Figure 2 is necessary, to demonstrate in a single overview how AWS time series have differing lengths and completeness.**

Section 2.2.2 and 2.2.3: maybe these two sections could be merged to improve the readability of the paper. Also the title should include the ERA-Interim product.

**Reply: The title of section 2.2.2 is changed "2.2.2 ERA-Interim and ERA5" as requested. But we prefer not to merge Sections 2.2.2 and 2.2.3, because the former describes Global reanalysis products whereas the latter describes Regional climate model products.**

L185-186: convoluted sentence, readability could be improved.

**Reply: The sentence is corrected as requested.**

The Surface Energy Balance (SEB) model uses AWS data as input. It iteratively solves for the value of Ts for which the energy budget is closed.

L192: what does the author mean exactly with the surface value of the calculated subsurface heat flux?

**Reply: We reformulated as follows:**

**...,** G is the subsurface heat flux, evaluated at the surface, and**...**

L215-217: the process of Smeets and van den Broeke., (2008) could be better explained or simply skipped (e.g. ... following the study of Smeets and van den Broeke., (2008) a value of ...).

**Reply: We simplified this paragraph as requested.**

Following the study of *Smeets and van den Broeke., (2008)* a $z0$ value of $1.3 *10^{-3}$ m

is chosen for S5, S6, and KAN_L when ice is at the surface, and $1.3* 10^{-4}$ m when snow covers the surface at these AWS sites.

L226-233: how is the subsurface part of the model initialized?

**Reply: We add more information about the initialization as follows:**

The subsurface model is initialized using measured density and temperature profiles at the date of station installation, and assuming no liquid water.

Figure 4: any comment on the fact that it appears that modeled surface temperature is 0C much more often than the observed, this could mean that modeled melt is overestimated.

**Reply: The extent to which this holds is hard to say, because both in model as in observations temperatures are capped at the melting point. Considerable uncertainty also exists in the 'observed' surface temperature. Given that observed and modelled ice melt agree well, a systematic bias in calculated melt is not supported.**

L251-252: I find this sentence a bit misleading, there is still information to be retrieved from 0C surface temperature (e.g. see previous comment about Figure 4), however it is true that the amount of melt cannot be assessed just by using melting surface temperature as a proxy.

**Reply: To clarify, we reformulated as follows:**

When temperature reaches the melting point, it no longer varies in time and as such it can no longer be used to evaluate SEB model performance. Instead, we assess...

Figure5: modeled data and observed data axis are inverted compared to Figure 4, this should be avoided at all cost. Be consistent with the chosen convention (e.g. modeled data always on the x-axis).

**Reply: Here Figure 5 is fixed to keep modeled data always on the x-axis.**

[Figure]

**New Figure 5**

L275-279: such generalization should be avoided in my opinion when site characteristics vary so much. E.g. at S5 the radiation penetration effect on total cumulative melt flux is neglectable but what about at a much higher elevation site like S10 where the melt flux is much smaller?

**Reply: Here we add another study about Greenland summit to give a radiation penetration range from lower station to the summit for the model uncertainty analysis.**

*Van den Broeke et al. (2008b) and Kuipers Munneke et al., (2009)* used a spectral albedo model based on the parameterization by *Brandt and Warren (1993)* to calculate subsurface penetration of shortwave radiation at S5 and at Greenland Summit station. Subsurface melt was only found to be important at S5, but with little influence on the total melt. Based on these results, here we assume that that neglecting subsurface radiation penetration in the SEB calculations has little effect on the total cumulative melt flux.

Section 4.1.1 and Figure 6: why there are no model values of surface height change for KANU and S10? Also doesn't the model simulate accumulation?

**Reply:**

**As the SEB model does not simulate accumulation, there are no model values of**

**surface height change for KANU and S10 which are situated above the equilibrium line.**

Why are measured height changes compared to modelled ice melt in Figure 6?

**Reply:**

**As we don't know the density of snow and firn which have melt, the SEB model present the modeled ice melt by assuming an ice density of 910 kg/m³. That's the reasons of measured height changes compared to modeled ice melt in Figure 6.**

Also some of the dashed lines are not continuous (e.g. S6, KANL, ...) why is this the case? A better explanation should be given here. (The nature of this comment is similar to my general comment about RACMO melt flux validation)

**Reply:**

**The dashed lines are not continuous due to the gap data of the model input, as shown in Figure 2.**

L309-311: reference about the cloud cover product used here?

**Reply: Here we use cloud cover data estimated from PROMICE AWS based on $L_{in}$ and air temperature. We added a reference for this equation.**

Probably owing to more frequent and thicker clouds along the T-transect (cloud cover 0.51 at KAN_L vs. 0.56 at THU_L in summer, using cloud cover estimates from PROMICE AWS based on Lin and air temperature according to (*Favier et al.*, 2004).

Figure8: I would rather put T2m and q2m on the same subplot (but different axis) since they are correlated rather than q2m with the wind.

**Reply: Combining your suggestion with the suggestion from the first reviewer, we decided to separate the plots of the three variables. If q2m and t2m are plotted on the same subplot, it is difficult to distinguish the magnitude of the amplitude in q2m.**

[Figure]

**New Figure 8**

Figure 11: this figure needs a bit of work from the reader to be fully understood. Providing additional descriptive text (e.g. at L470) to assist the reader would help. Also consider keeping the y-axis symmetric and with the same range. This would help in assessing the difference between different fluxes.

**Reply:**

**We now added the slope unit for every subplot to assist the reader. But if we keep the y-axis range for (c), (d), (e) and (f) same with Fig11 (a) (-15 ~25 W m$^{-2}$/$\sigma_{GBI,NAO}$), then it will difficult to distinguish (d), (e) and (f), because the ranges of (d), (e) and (f) are -4~6 W m$^{-2}$/$\sigma_{GBI,NAO}$. As a compromise, we decided to make Fig11(d), Fig11(e) and Fig11(f) with the same y-axis range instead.**

[Figure]

**New Figure 11**

L512-515: convoluted sentence, readability could be improved.

**Reply: we changed this sentence as requested.**

Next we discuss the spatially different response of western GrIS climate and melt to GBI. To that end, Fig. 12 shows maps of the JJA GBI dependency for temperature (Fig.12a) and melt (Fig. 12c) for Greenland and its immediate surroundings using RACMO2. Fig. 13a shows the regional 500 hPa height anomaly from ERA5 associated with variations in GBI.

Figure 12 and 13: missing subplot titles and colorbar labels.

**Reply: Combined with the first reviewer's suggestions with the "diverging color ramp", we have changed Figures 12 and 13 as requested.**

[Figure]

**New Figure 12**

[Figure]

**New Figure 13**

**References**

**Reply: We have added these references.**

Charalampidis, C., van As, D., Box, J. E., van den Broeke, M. R., Colgan, L. T., Doyle, S. H., et al. (2015). Changing surface-atmosphere energy exchange and refreezing capacity of the lower accumulation area, West Greenland. The Cryosphere, 9(6), 2163-2181.

Vandecrux, B., Fausto, R. S., Langen, P. L., van As, D., MacFerrin, M., Colgan, W. T., et al. (2018). Drivers of firn density on the Greenland ice sheet revealed by weather station observations and modeling. Journal of Geophysical Research: Earth Surface, 123, 2563-2576.

Favier, V., P. Wagnon, J.P. Chazarin, L. Maisincho, and A. Coudrain (2004), One-year measurements of surface heat budget on the ablation zone of Antizana Glacier 15, Ecuadorian Andes, Journal of Geophysical Research, 109(D18), doi:10.1029/2003 jd004359.

Kuipers Munneke, P., van den Broeke, M. R., Reijmer, C. H., Helsen, M. M., Boot, W., Schneebeli, M., and Steffen, K.: The role of radiation penetration in the energy budget of the snowpack at Summit, Greenland, The Cryosphere, 3, 155-165, https://doi.org/10.5194/tc-3-155-2009, 2009.

---

## Referee Report (RR1)

The revisions made by the authors highly improved the manuscript. Most, if not all, the comments raised by the two reviewers have been addressed adequately making the paper mostly suitable for publication.

Despite this there is one aspect of my initial review which hasn't been completely addressed as I hoped. Being this one of the main issues I had with manuscript I feel the need to point out the concerns I have again. The aspect I am referring to is the point 2 "Validation of RACMO melt flux" of my initial review. For reference I am reporting here my initial comment as well as how it was addressed by the authors:
* * *
**2. Validation of RACMO melt flux:**

This is probably the main problem the paper has in its current version. Nowhere in the manuscript I have found a validation of RACMO modeled melt flux, yet this variable is used extensively in the discussion section. In line 449 the reader is referred to Noël et al., (2018),whose SEB evaluation at sites S5, S6, S9, and S10 includes the melt flux as well (e.g. Tables 2-5). Ultimately modeling melt accurately is one of the final goals of any surface energy balance model and regional climate model used to study ice sheets changes. An often proposed and used approach to this type of validation is to evaluate point studies (e.g. like the SEB modeling results here presented) against in-situ observations and then evaluate regional climate models against the point studies. The current manuscript already uses this framework for all the SEB components, but in my opinion it needs to include the same analysis for the melt flux as well.

**Reply:**

(1) A validation of SEB model against in-situ ablation rate observations is presented in Figure 5 (Average 10-day modeled and observed ice melt). This only works for ice with known density, i.e. at KAN_L, KAN_M, S5, S6 and THU_L which are situated below the equilibrium line. See for the detailed descriptions Section 3.2 (SEB model evaluation) and discussion of Figure 5.

(2) We added a row in the evaluation tables in which RACMO2.3p2 melt rate is compared to that from the SEB model for station KAN_L, KAN_M, KAN_U, THU_L and THU_U. Previously, an extensive evaluation of RACO2.3p2 ablation rate with all available observations from Greenland S5, S6, S9 and S10 was performed by Noël et al., (2018) and also showed good agreement.
* * *
While the authors added the comparison of melt rates from RACMO and the SEB model (e.g. reply (2)), a proper validation of the SEB model at sites above the equilibrium line is not exhaustively addressed in my opinion. I do understand that this type validation is more challenging due to not knowing the density of snow and firn, as the authors properly comment later on in their reply to my review. Despite this, if I understand correctly, the SEB model used

has a subsurface module based on SOMARS (e.g. L241) which, if I am not wrong, should simulate, among other variables, snow and firn density. This can be used to compare observations of e.g. relative surface height change (which are recorded by AWS) with simulated values of melt, as it is done for sites below the equilibrium line. If for some reasons this is not possible and not presented in the manuscript, conclusions regarding these sites (e.g. accumulation zone of the ice sheet) should be presented in a more careful manner. The ablation zone and the accumulation zone highly differ in how they are affected by the surface processes described by SEB models and I think we cannot take for granted that models that work well in one region work well also in the other without proper ground truth validation.

---

## Author Response (AR2)

**We thank the reviewer for the comments, which have further improved the paper.**

**Reviewer comment 2. Validation of RACMO melt flux:**

While the authors added the comparison of melt rates from RACMO and the SEB model (e.g. reply (2)), a proper validation of the SEB model at sites above the equilibrium line is not exhaustively addressed in my opinion. I do understand that this type validation is more challenging due to not knowing the density of snow and firn, as the authors properly comment later on in their reply to my review. Despite this, if I understand correctly, the SEB model used has a subsurface module based on SOMARS (e.g. L241) which, if I am not wrong, should simulate, among other variables, snow and firn density. This can be used to compare observations of e.g. relative surface height change (which are recorded by AWS) with simulated values of melt, as it is done for sites below the equilibrium line. If for some reasons this is not possible and not presented in the manuscript, conclusions regarding these sites (e.g. accumulation zone of the ice sheet) should be presented in a more careful manner. The ablation zone and the accumulation zone highly differ in how they are affected by the surface processes described by SEB models and I think we cannot take for granted that models that work well in one region work well also in the other without proper ground truth validation.

**Reply:**

(1) We agree with the reviewer that evaluating melt energy in the accumulation zone is highly important, but there currently is a lack of suitable observations, and it will require enhanced observational efforts to achieve this. The reason is that melt energy in the accumulation zone cannot be reliably assessed from stake or sonic height ranger observations, because vertical motion of the snow surface can be caused by multiple processes: changing stake/AWS base depth, differential firn compaction between the stake/AWS base and the surface, and surface mass balance processes that include melt but also e.g. erosion by drifting snow. Because at the same time, the melt fluxes away from the ice margins are relatively small, these processes significantly decrease the signal to noise ratio in the accumulation zone. So even if the density of the layer that has been removed would be perfectly known, which is almost never the case, this cannot be one-on-one converted into a melt flux.

(2) For the reasons outlined above, we adopted a comparison with AWS-derived melt energy from solving the surface energy balance, which is acceptable given its relatively high accuracy. But this can only be done if the AWS measure a reliable radiation balance. For that reason, GC-Net stations, which cover most of the

accumulation zone, cannot be used, and we are limited to a comparison with the higher PROMICE stations in west Greenland. We therefore agree with the reviewer that the resulting scarcity of evaluation points in the accumulation zone warrants caution when discussing these results.

(3) We added these considerations to the manuscript (lines 270-284), marked in red:

**In the accumulation zone, vertical motion of the snow surface can be caused by several processes: changing stake/AWS base depth, differential firn compaction between the stake/AWS base and the surface, and surface mass balance processes that include melt but also e.g. erosion by drifting snow. Because at the same time, the melt fluxes away from the ice margins are relatively small, these processes significantly decrease the signal to noise ratio in the accumulation zone. So even if the density of the layer that has been removed would be perfectly known (which is almost never the case), this cannot be one-on-one converted into a melt flux. For these reasons, modelled melt rate in the accumulation zone is usually evaluated by comparing it to the melt energy obtained from AWS observations. However, this can only be done if the AWS measure a reliable radiation balance, which limits the effort to the higher PROMICE stations in west Greenland. The resulting scarcity of evaluation points in the accumulation zone warrants caution when interpreting the variability of melt rates in the Greenland interior as presented in this paper.**